# Neurobiological Mechanisms Underlying Psychological Dysfunction After Brain Injuries

**DOI:** 10.3390/cells14020074

**Published:** 2025-01-08

**Authors:** Prashin Unadkat, Tania Rebeiz, Erum Ajmal, Vincent De Souza, Angela Xia, Julia Jinu, Keren Powell, Chunyan Li

**Affiliations:** 1Translational Brain Research Laboratory, The Feinstein Institutes for Medical Research, Manhasset, NY 11030, USA; 2Department of Neurosurgery, North Shore University Hospital at Northwell Health, Manhasset, NY 11030, USA; 3SUNY Downstate College of Medicine, Brooklyn, NY 11225, USA; 4Department of Neurosurgery, Staten Island University Hospital at Northwell Health, Staten Island, NY 10305, USA; 5Biology Department, Adelphi University, Garden City, NY 11530, USA; 6Department of Neurosurgery, Donald and Barbara Zucker School of Medicine at Hofstra/Northwell, Hempstead, NY 11549, USA

**Keywords:** psychological dysfunction, anxiety, depression, brain injury, cerebral ischemia, dementia

## Abstract

Despite the presentation of similar psychological symptoms, psychological dysfunction secondary to brain injury exhibits markedly lower treatment efficacy compared to injury-independent psychological dysfunction. This gap remains evident, despite extensive research efforts. This review integrates clinical and preclinical evidence to provide a comprehensive overview of the neurobiological mechanisms underlying neuropsychological disorders, focusing on the role of key brain regions in emotional regulation across various forms of brain injuries. It examines therapeutic interventions and mechanistic targets, with the primary goal of identifying pathways for targeted treatments. The review highlights promising therapeutic avenues for addressing injury-associated psychological dysfunction, emphasizing Nrf2, neuropeptides, and nonpharmacological therapies as multi-mechanistic interventions capable of modulating upstream mediators to address the complex interplay of factors underlying psychological dysfunction in brain injury. Additionally, it identifies sexually dimorphic factors as potential areas for further exploration and advocates for detailed investigations into sex-specific patterns to uncover additional contributors to these disorders. Furthermore, it underscores significant gaps, particularly the inadequate consideration of interactions among causal factors, environmental influences, and individual susceptibilities. By addressing these gaps, this review provides new insights and calls for a paradigm shift toward a more context-specific and integrative approach to developing targeted therapies for psychological dysfunction following brain injuries.

## 1. Introduction

Psychological dysfunction, manifesting as a complex spectrum of affective disorders, imposes a substantial global health burden [1,2,3,4,5]. Affecting an estimated 5–15% of the global adult population, the development of effective therapeutic strategies for psychological dysfunction is a critical research priority. This need is particularly urgent for individuals with acquired brain injury, who exhibit notably higher prevalence rates of psychological dysfunction, estimated at 30–50% [6,7,8,9,10,11,12]. Notably, injury-dependent psychological dysfunction demonstrates lower responsiveness to pharmacotherapy, with reported efficacy rates of 30–40%, compared to 60–70% efficacy in injury-independent contexts [1,5,13,14,15,16] (Figure 1). Despite substantial research efforts and the significant societal impact of this condition, the mechanisms contributing to the lower treatment efficacy in injury-related psychological dysfunction, compared to injury-independent cases, remain insufficiently understood, highlighting the pressing need for innovative, targeted therapeutic strategies.

Two of the most common forms of affective disorder, anxiety and depression [1,2,3,4,5], are exacerbated both by brain injury and the psycho-emotional disturbances related to the acute phase of the inciting event [13,14]. The onset of these conditions reduces patient engagement in rehabilitation, impairing recovery from the brain injury and subsequently escalating psychological dysfunction, creating a self-perpetuating cycle. Therapeutic interventions are further challenged by the heterogeneity of psychological dysfunction presentations in clinical settings [17,18,19,20,21,22,23,24]. Meta-analyses on injury-dependent psychological dysfunction reveal a broad range of symptom severity, from mild to severe, regardless of the initial injury severity, influenced by various contributing factors. Research into injury-dependent psychological dysfunction has often mirrored studies on injury-independent psychological dysfunction, potentially overlooking etiologically specific factors that may contribute to reduced treatment efficacy. This gap highlights the need for more effective strategies to manage established psychological dysfunction following brain injury. It is crucial to re-evaluate the current neurobiological mechanisms targeted for therapeutic interventions addressing psychological dysfunction in various brain injury models, to develop more effective treatment strategies that enhance clinical outcomes.

This review aims to identify shared features and distinctions between injury-independent and injury-dependent psychological dysfunction, with the goal of pinpointing promising research directions specific to brain injury contexts. It assesses psychological dysfunction arising from ischemic stroke [8,10], intracerebral hemorrhage (ICH) [8], subarachnoid hemorrhage (SAH) [7], traumatic brain injury (TBI) [6,9], and vascular cognitive impairment and dementia (VCID) due to chronic cerebral hypoperfusion (CCH) or as a long-term consequence of acute brain injury [25], as well as distress observed in non-injury populations. Currently, there is no evidence suggesting fundamental differences in the mechanisms underlying psychological dysfunction across specific brain injuries [26]. However, distinct etiologies and variable damage progression patterns across these conditions indicate the potential for unique pathways influencing the development of psychological dysfunction in each injury type (Figure 2). We evaluated psychological dysfunction across diverse contexts to address three key aims: (1) to assess whether different types of brain injury influence psychological health in distinct ways, (2) to identify potential variations in targeted neurobiological mechanisms and therapeutic strategies between injury-independent and injury-dependent contexts, and (3) to evaluate the applicability of findings from non-injury (“healthy”) populations to those with brain injuries. By highlighting these aspects, this review seeks to propose innovative avenues for investigating psychological dysfunction secondary to brain injury, ultimately aiming to advance therapeutic interventions.

## 2. Neurobiological Mechanisms and Therapeutic Strategies for Psychological Dysfunction in Injury-Independent and Injury-Dependent Contexts

To investigate the observed disparities in treatment efficacy between injury-independent and injury-dependent psychological dysfunction, we analyzed current trends in the neurobiological mechanisms explored in various brain injuries, the therapeutic interventions employed and their respective efficacies, and the brain sub-regions studied. Additionally, we compared the neurobiological mechanisms across different types of brain injuries to assess whether etiological variations should be considered in the treatment of psychological dysfunction. Overall, the mechanisms examined do not appear to be context-specific, with common themes including neuroinflammation, immunoregulation, apoptosis, oxidative stress, neurotransmission, synaptic plasticity, stress and sex hormones, amyloid and insulin metabolism, neuropeptides, and methylation. Similarly, the brain sub-regions investigated are consistent across both injured and non-injured contexts, including the hippocampus, cortex, amygdala, hypothalamus, white matter, striatum, and ventral tegmental area (VTA) (Figure 3). In injury-dependent psychological dysfunction, the primary symptoms explored include anxiety, depression, and fear responses (Table 1). This review, therefore, investigates generalized anxiety disorder (GAD), major depressive disorder (MDD), and panic disorder as examples of injury-independent psychological dysfunction, with a particular focus on comparing the neurobiological mechanisms and therapeutic interventions used in both injury-dependent and injury-independent contexts.

### 2.1. Psychological Dysfunction Manifesting in the Absence of Apparent Brain Injury

Previous reviews have explored the various underlying factors that influence the clinical presentation of generalized anxiety disorder, major depressive disorder, and panic disorder [27,28,29,30]. In this review, we provide a succinct overview of these factors, including neurobiological mechanisms, the roles of specific brain regions, and therapeutic interventions. This framework is used to assess whether neurobiological mechanisms and responses to therapeutic interventions differ between injured and non-injured contexts, aiming to more effectively identify potential targets for therapeutic intervention.

#### 2.1.1. Neurobiological Mechanisms

The primary neurobiological mechanisms underlying psychological dysfunction in the absence of brain injury include neuroinflammation, which is partially influenced by stress-induced alterations in immune regulation, along with oxidative stress, DNA methylation, dysregulation of neurotransmitters and neuropeptides, disruptions in synaptic plasticity, and imbalances in sex and stress hormones. A primary theory of inflammation-mediated psychological dysfunction dictates that emotional stress can trigger peripheral immune systems, causing the release of pro-inflammatory cytokines [31]. The hypothalamic–pituitary–adrenal (HPA) axis and the sympathetic–adrenal–medullary system are activated by stress, which mediates the release of corticotropin-releasing hormone (CRH) levels by the hypothalamus, and the subsequent release of inflammatory cytokines, such as TNF-α, IL-1β, and IL-6, and chemokines, such as CXCL12. This affects the presence of serotonin, GABA, and glutamate, thus creating neurotransmitter dysregulation and impairing neurogenesis. The prolonged microglial and astrocytic activation also has an effect on oxidative stress, engendering an increase in reactive oxygen and nitrogen species [31]. The increase in reactive oxygen species (ROS) is accompanied by high levels of lipid peroxidation within the brain, as well as cyclooxygenase-2 (COX-2) activity [28,29]. This results in overexpression of the glutathione redox system and further activation of the HPA axis, forming a self-reinforcing feedback loop that promotes the production of neurotoxic substances and, thus, anxiety and depression. Hypersensitization of the HPA axis is a defining characteristic of anxiety, with stress producing an increase in plasma cortisol but not adrenocorticotropic hormone (ACTH) [32]. Depression, however, occurs with higher cerebrospinal concentrations of corticotropin-releasing factor (CRF) and a blunted ACTH response to CRF injections, suggesting the presence of sensitized glucocorticoid receptors secondary to chronic CRF elevations. Dysregulation of the stress response may be sexually mediated, with females exhibiting heightened sensitivity to CRF within the locus coeruleus [33,34,35], while estrogen fluctuations can increase depression development in females [36].

In addition to the interactions between inflammation, oxidative stress, and hormones, neurotransmitter and neuropeptide dysregulation, synaptic plasticity disruption, and DNA methylation can all interact to modulate psychological dysfunction. A fundamental aspect of psychological dysfunction is neurotransmitter disruption, which occurs with such a high propensity that a majority of pharmacological interventions focus on targeting it [30,31,32]. Disruptions in the GABA system are linked to both anxiety and panic disorder, as are disturbances in serotonin circuitry, as are disruptions in the reactions to cortisol. Depression in particular is classically associated with deficiencies in serotonin, noradrenaline, and dopamine, alongside disruptions in glutamatergic and GABAergic signaling [37]. Monoaminergic neurotransmitter disruption contributes to changes in synaptic plasticity, characterized by the alteration in neurotrophin levels, particularly those of brain-derived neurotrophic factor (BDNF). This contributes to reductions in ERK and the activity of AkT, which contributes to impairments in neuronal plasticity and survival. Concurrently, disruptions in neuropeptide expression and activity are observed in psychological dysfunction [32]. This can include hypersensitivity to cholecystokinin (CCK), the most abundant neuropeptide within the brain, decreased expression of neuropeptide Y (NPY), chronic galanin overexpression, decreased oxytocin, and increased vasopressin. These are primarily related to chronic dysregulation of the stress and pain response system, leading to emotional disturbance. Finally, DNA methylation may also contribute to observed symptomology in injury-independent psychological dysfunction [27]. Assessments have indicated that hypermethylation of BDNF and NR3C1 is associated with an increased risk of depression [38], while the methylation profiles of DNMT1/3A, EZH2, and IL-6 correlate to the severity of anxiety [39].

#### 2.1.2. The Impact of Specific Brain Regions on Neurobiological Mechanisms

The neurobiological changes discussed earlier can occur throughout the brain; however, certain regions exhibit a stronger association with specific mechanisms. The hippocampus, which is integral to mood regulation and defensive responses, plays a crucial role in the onset of anxiety and depression [40]. In the context of psychological dysfunction, the hippocampus has been shown to undergo significant volume reduction, along with dysregulation of neurogenesis, synaptic plasticity, neurotransmission, neuroinflammation, and stress hormone activity in various studies [40,41,42,43,44]. Evidence suggests that the volume of the left CA1 region of the hippocampus can independently predict the duration of illness, while the CA2 through CA4 subregions are the only ones to show reductions in first-onset major depressive disorder, highlighting the hippocampus’s critical role in psychological dysfunction [44]. The cortex, particularly the prefrontal cortex, demonstrates dysfunction associated with psychological dysfunction [45,46,47,48]. The impact of this dysfunction is intricately modulated by the prefrontal cortex’s connectivity with other brain regions. Specifically, in primates, including humans, the specialized lateral, medial, and orbital sectors of the prefrontal cortex are linked to the amygdala and brainstem—autonomic structures that influence emotional and physiological arousal through the limbic cortex [46].

The amygdala, recognized as a central hub in the brain’s stress circuitry, is another critical brain region involved in psychological dysfunction [49]. It has been proposed that the amygdala may not function as a distinct entity but rather as a network of extensions connecting to the cortices, striatum, hypothalamus, and brainstem. Within the amygdala, several neurotransmitters, including serotonin, GABA, and dopamine, have been implicated in the onset of psychological dysfunction, supporting the idea of the amygdala as a key independent regulator in emotional processing.

The hypothalamus, a key component of the hypothalamic–pituitary–adrenal (HPA) axis that regulates the body’s physiological stress response, plays a crucial role in psychological dysfunction. In depression, increased hypothalamic volume has been correlated with severity [50], while dysregulation of cortisol, corticotropin-releasing hormone (CRH), and oxytocin has been linked to generalized anxiety disorder [51]. The ventral tegmental area (VTA) is another significant region, where the activity of tyrosine kinase [52] and dopamine [53] is involved in regulating depression and anxiety, respectively. Similar to the hypothalamus, individuals with anxiety and depression exhibit larger VTA volumes [54]. These regions represent only a subset of the mechanisms underlying injury-independent psychological dysfunction but highlight the interconnected pathways that contribute to the presentation of GAD, major depressive disorder, and panic disorder.

#### 2.1.3. Therapeutic Interventions

Given the high prevalence of psychological disorders in the general population and the significant societal burden they impose, along with the complex mechanisms and pathways underlying their symptomatology, numerous therapeutic interventions have been evaluated [55,56,57,58,59]. These interventions can be broadly categorized into pharmacological treatments, such as benzodiazepines, selective serotonin reuptake inhibitors (SSRIs), serotonin–norepinephrine reuptake inhibitors (SNRIs), non-benzodiazepine anxiolytics, monoamine oxidase inhibitors (MAOIs), and tricyclic antidepressants (TCAs), as well as psychological therapies like cognitive behavioral therapy (CBT) and acceptance and commitment therapy (ACT). Both benzodiazepine and non-benzodiazepine anxiolytics act on the binding of the neurotransmitter gamma-aminobutyric acid (GABA), enhancing its inhibitory effects, thus calming neuronal activity and reducing anxiety [60]. SSRIs and SNRIs, on the other hand, function by preventing the reuptake of serotonin and norepinephrine, thereby increasing the relative bioavailability [61], while MAOIs block the monoamine oxidase enzyme, preventing the breakdown of neurotransmitters in the brain [62]. On the whole, the pharmacological interventions can be classified as functioning via modulation of neurotransmission within the brain. Additional non-pharmacological approaches, including physical exercise and neuromodulation, target other neurobiological mechanisms, including neurotransmission, neuroplasticity, inflammation, oxidative stress, hormonal dysregulation, peptides, and the endocrine system [56,58,63]. Despite the variety of available treatment options, a subset of individuals remains unresponsive to these therapies. Specifically, approximately 30% of individuals with major depressive disorder exhibit treatment resistance [16], and about 40% of individuals with anxiety disorders are similarly resistant to treatment [64]. However, as we will demonstrate, treatment resistance is notably higher in injury-dependent psychological dysfunction.

**Table 1 cells-14-00074-t001:** Neurobiological mechanisms underlying anxiety and depression across various types of brain injuries, as investigated through animal models. (↑ = increase; ↓ = decrease; → = leads to).

Mechanism	Trends	Sex-Specificity	Brain Region Specificity	Disease/Model	Ref.
Apoptosis	P38-MAPK, ↓ERK, Akt/mTOR → ↓neurons → anxiety, compulsive-like behaviors, and decreased activity levels.	N/A	ERK, SAH: Hypothalamus Parietal–entorhinal cortex	SAH Ischemic stroke	[65,66,67,68]
	↑Caspase 3/8/9 cleavage → ↓neurons → depressive behaviors	N/A	Hypothalamus Prefrontal cortex Hippocampus	Global ischemia	[69]
	P2Y14r → ↑p54, p46, BAX, caspase 3 cleavage → depressive behaviors	N/A	Cortex	SAH	[70]
	Apoptosis-inducing factor, Cytochrome C → depressive behaviors	N/A	Ipsilateral hippocampus	mTBI	[71]
Neuroinflammation	↑TNF-α, IL-1B, IL-6, and NF-κB → Anxiety and depression	ICH: Estrogen modulates astrocytic-mediated inflammation and the presentation of depression	N/A	TBI ICH	[17,72,73]
	Astrocytosis, microgliosis → Anxiety and depression		ICH (depression): White matter Hippocampus Striatum ICH (anxiety and depression): Ventral tegmental area	ICH TBI Stroke	[17,24,73,74,75,76]
	↓pTRkB, ↓pERK, ↓pCREB → Anxiety	N/A	Hippocampus	CCH + CRS	[77]
	↓PKA/CREB → Anxiety	N/A	Amygdala	TBI	[78]
	HDAC3 → NFkB → p65 → Cox1 → Anxiety	N/A	Cortex	Photothrombotic stroke	[79]
	↓Pink1 → inflammation → Anxiety and depression	N/A	Ipsilateral hemisphere	ICH	[80]
	↓Nrf2, TrkB-BDNF, and Trbk-PI3K; ↑Nrf2-Keap1-p62 → Anxiety and depression	N/A	N/A	ICH	[81,82,83]
Immunoregulatory Dysfunction	↑pPERK → ↑STING → ↑IFNβ → Th1 → Anxiety and depression	N/A	N/A	TBI	[84]
	↑Neutrophil elastase → ↓sepina3n → Anxiety	N/A	Cortex	TBI	[85]
	FOXO1 → VCAN, BAX, ↓ neutrophils → IL-6, ferroptosis → Depression	N/A	N/A	TBI	[86]
Oxidative Stress	↓SOD, GSH, CAT → Anxiety ↓SOD → Depression	ICH: Estrogen modulates SOD and the presentation of depression	N/A	ICH CCH	[73,74,87,88]
	↓Glutathione metabolism → Anxiety	N/A	Cortex	TBI	[89]
	HDAC1, HDAC3, NOX4 → prostaglandin → Anxiety	N/A	Cortex	TBI	[90]
	↑Lipid peroxidation → Anxiety and depression	N/A	N/A	ICH CCH TBI	[79]
	Keap1-Nrf2-p62 modulates oxidative stress to regulate depression	N/A	N/A	Stroke	[72,82,87]
Neurotransmission	↓Dopaminergic neurons → depression	N/A	N/A	ICH	[52]
	↓α-tubulin → ↓microtubule stability → ↓dopaminergic neurons → depression	N/A	Ventral tegmental area	ICH	[76]
	↑Dopamine → Anxiety	N/A	Striatum	CCH	[91]
	↓GABA_B2_ subunits → Anxiety ↓GABA_B1_, GABA_B2_ → Depression ↓GABAr → Anxiety	N/A	CCH: (GABA_B2_ subunits: Amygdala GABA_B1_, GABA_B2_: Hippocampus) TBI: Basolateral amygdala	CCH TBI	[21,22,92]
	Dysfunctional glutamate uptake and ↑glutamate → depression ↑glutamate → Anxiety, depression	N/A	SAH/ICH: Striatum TBI: central amygdala	SAH ICH TBI	[74,81,93,94,95,96,97]
	NMDAR → Anxiety and depression	N/A	Hippocampus	TBI: anxiety Stroke: depression	[98,99,100]
	↑Kynurenine → ↓serotonin → Depression	N/A	ICH: Perihematomal region CCH, Stroke: hippocampus	ICH CCH Stroke	[101,102,103]
	↓Nrf2/BDNF → Anxiety and depression	N/A	ICH: Perihematomal region	ICH	[81,83]
	↓BDNF → ↓TrkB → ↓ NCAM → GIRK, Kir3 → Anxiety and depression	N/A	CCH: Amygdala	CCH	[21,104]
	↑Acetylcholine → Anxiety	N/A	Basolateral amygdala	TBI	[92]
	↑HCN1 and ↓KCNQ3 → Anxiety and depression	N/A	Amygdala	CCH	[104]
	↓CNR1, ↓COMT, ↓VEGF2r → Anxiety	N/A	Cortex	TBI	[105]
	↓CB1 receptor → Depression	N/A	Ventromedial hypothalamus	Stroke	[106]
Synaptic Plasticity	↑Lipid peroxidation → ↓BST, LTP, PPR → Anxiety	N/A	Hippocampus	CCH	[107]
	↓ mRNA β1-catenin, ↓TGF-β1, ↓NR2B → Anxiety	N/A	Hippocampus	CCH	[108]
	↑GSK-3β → Anxiety	N/A	Hippocampus	CCH	[108]
	↓BDNF → ↓PSD95 → Depression	N/A	Hippocampus	CCH	[109]
	↓PSD95, ↓SYN, ↓GAP43, SYP → Anxiety	N/A	Hippocampus	CCH + CRS	[77]
	NR2A, NR2B, NMDA proteins, GAD67, GAD65, GluA2 → Anxiety	N/A	Dorsal hippocampus Ventral hippocampus Basolateral amygdala	TBI	[110]
	↓CNPase, ↓synaptophysin → Anxiety and depression	N/A	White matter	CCH	[111]
Stress Hormones	Glucocorticoid receptor translocation → Depression	N/A	Hippocampus	CCH	[112]
	↑CRF-1, ↑CRF-1R → Anxiety	N/A	N/A	mTBI	[113]
	↑CRH1, ↑CRHR1 → ↑BDNF, ↑TrkB, and ↑pCREB-ir → Anxiety	N/A	Amygdala Hippocampus	Transient global ischemia	[114,115,116]
	↑CRHR1 → p62 → synaptic loss → Depression	N/A	Hippocampus	tMCAO	[82]
	CRFR2 → Anxiety	Females: ↓CRFR2 in the DH Males: ↑CRFR2 in VH, ↓ CRFR2 in amygdala	Ventral hippocampus Dorsal hippocampus Amygdala	TBI	[117]
	↑NGF → Anxiety	N/A	N/A	Transient global ischemia	[118]
Sex Hormones	Estrogen, progesterone, testosterone → α-1 GABA_A_, ↓Dopamine → Anxiety	Diestrus and proestrus phases in females affect severity	N/A	mTBI	[19,119,120]
	Estrogen, testosterone → SERT→ serotonin → Anxiety	Proestrus females experience increased serotonin	N/A	TBI	[119,121]
Amyloid/Insulin Metabolism	↑BACE → ↑AAP → ↑Aβ → Anxiety	N/A	Hippocampus	CCH	[122,123]
	GlcNAc dysfunction, ↓Insulin signaling pathway → Anxiety	N/A	N/A	CCH	[124]
Trophic Factors	↓BDNF → ↓Neurogenesis → Depression	N/A	Hippocampus Striatum	Stroke	[125]
	↓VEGF → dopamine → Anxiety	N/A	Pre-frontal cortex	TBI	[126]
Neuropeptides	CGRP → Nrf2, eNOS → Anxiety and depression	The presentation of decreased anxiety and depression in females is mediated by CGRP	Amygdala Thalamus Hippocampus White matter Pericontusional cortex	TBI	[127]
	↑ACE/Ang II/AT1R → ↓CBF, ↑Oxidative stress, ↑Neuroinflammation → Anxiety	N/A	Hippocampus	CCH	[128]
Methylation	↓m6A demethylase, ↓FTO → Depression	N/A	N/A	Stroke	[129]
	↑DNMT, ↑HDAC2 → GADD45 damage → Anxiety	N/A	Amygdala	TBI	[130,131]

### 2.2. Traumatic Brain Injury

#### 2.2.1. Clinical Presentation and Clinical Therapeutic Efficacy

TBI affects over 25 million individuals annually [132], with its causes encompassing a wide range of events, including military combat, vehicular accidents, recreational or sports-related injuries, and falls, although mild TBI (mTBI) is often underreported [133]. TBI is defined as a disruption in brain structure or function resulting from an external force, typically acute in nature. The causes of TBI lead to varying degrees of injury, classified as mild, moderate, or severe, with a corresponding spectrum of symptoms. Individuals across this spectrum are at increased risk for developing major depressive disorder, posttraumatic stress disorder (PTSD), generalized anxiety disorder, and elevated suicidality, which significantly contributes to morbidity. In the absence of injury, approximately 5–15% of the general population is diagnosed with a DSM-IV Axis I disorder; however, within the first year following TBI, this rate increases to as high as 60% of individuals [1,2,3,4,5]. Treatment for individuals exhibiting depressive symptoms includes pharmacological interventions such as SSRIs, MAOIs, and TCAs, in addition to non-pharmacological approaches like CBT [134,135,136]. However, meta-analyses of randomized clinical trials suggest that current interventions show no significant benefit over placebo in cases of TBI-induced depression [134,136], with only certain dopaminergic agents demonstrating improvements in general behavioral outcomes [137].

#### 2.2.2. Neurobiological Mechanisms and Pre-Clinical Therapeutic Targets

A review of the literature reveals that psychological dysfunction following TBI is mediated through dysregulation of multiple biological pathways, including apoptosis, neuroinflammation, immunoregulatory dysfunction, oxidative stress, neurotransmission, synaptic plasticity, hormonal imbalances (stress and sex hormones), trophic factors, neuropeptides, and DNA methylation (Table 1 and Table 2). TBI is primarily characterized by an acute primary injury that results in a localized area of damage, followed by a secondary cascade of injury [138]. Mechanistic studies have investigated contributions across different brain regions. Although the initial TBI can lead to overt tissue damage, the role of apoptosis in psychological dysfunction remains underexplored. However, evidence suggests that apoptosis-inducing factors such as apoptosis-inducing factor and cytochrome C in the hippocampus play a role in modulating depressive behaviors following mild TBI [71]. Additionally, in the acute stage of TBI, elevation of forkhead box protein O1 (FOXO1) enhances the interaction between Versican (VCAN) and BCL-2-associated X protein (BAX), thereby inhibiting BAX translocation to the mitochondria and exerting anti-apoptotic effects [86]. This promotes neuronal degradation, later neuroinflammation, and depressive symptomology.

In contrast to the relatively sparse assessment of apoptosis’ effect on psychological dysfunction in TBI, neuroinflammation and immunoregulatory dysfunction, which are associated with the primary injury, have been more extensively studied. At the surface level, increased astrocytosis, microgliosis, and the release of inflammatory cytokines, including TNF-α, IL-1B, IL-6, and NF-κB, potentiate depressive and anxious symptomology [17,72]. This may be connected to the decrease in protein kinase A (PKA), leading to the decreased activation of cyclic AMP response element-binding protein (CREB), the modulation of which affects astrocyte function [72]. The observed anti-apoptotic effect of FOXO1 mediates the activation of neutrophils, resulting in an increase in IL-6 production [86]. It also promotes ferroptosis in neutrophils via the transferrin receptor (TFRC) mechanism. This disruption in iron homeostasis within oligodendrocytes leads to a reduction in myelin basic protein and contributes to depressive symptoms. Within neutrophils, the increased activation of neutrophil elastase also decreases serpina3n mRNA levels, impacting neuronal function and contributing to anxiety-like behavior [85]. Additionally, increased phosphorylation of extracellular signal-regulated kinase (pERK) leads to activation of the STING pathway, and increased interferon β [84]. This modulates the presentation of Th1 cells and the presentation of both anxiety and depression. Of the inflammatory effects, only neutrophil elastase [85] and PKA/CREB signaling [78] have been specifically linked to the cortex and amygdala, respectively. Other pathways have not been assigned a regional specificity.

Alongside apoptosis and inflammation, the primary injury also triggers immediate changes in oxidative stress, which contribute to psychological dysfunction. This is characterized by dysregulated interactions between glutathione and histone deacetylases (HDACs) [89,90] and increased lipid peroxidation [72]. Activation of HDAC1, HDAC3, and NADPH oxidase 4 (NOX4) [90] influences prostaglandin production in the cortex via microglia [78], which is associated with heightened anxiety and altered glutathione metabolism [89].

Following the primary injury, secondary damage is associated with disrupted neurotransmission and synaptic plasticity, including dysregulated expression of NMDAr and glutamatergic activity [99,110], reduced expression of GABAergic neurons [92], and decreased expression of VEGF receptor, cannabinoid receptors, and catechol-O-methyltransferase (*Comt*) [105]. The increase in NMDAr expression in the hyper-acute phase is followed by a decrease in the chronic phase, mediating the hyper-acute increase in Na^2+^ and Ca^2+^ influx into the cell [93,94,95,96,99,105]. This decreases brain-derived neurotrophic factor (BDNF) and nerve growth factor (NGF) production within the hippocampus and promotes the development of anxiety. The changes in NMDAr are further accompanied by bilateral alterations in synaptic glutamatergic a-amino-3-hydroxy5-methyl-4-isoxazolepropionic acid (AMPA; GluA1, GluA2) and GABA-synthetic (GAD65, GAD67) proteins, leading to alterations in synaptic plasticity [110]. Anxiety in TBI is also potentiated by the loss of GABAergic neurons and GABA_A_ receptor subunits in the basolateral amygdala (BLA), alongside an increase in acetylcholine receptors (α_7_-nAChR) [92]. This mediates both a reduction in inhibition within the BLA, as well as an increase in the excitability of principal neurons within the system. Catecholaminergic and endocannabinoid dysregulation also underlies the presentation of psychological dysfunction in TBI [105]. The decrease in *Comt* in the prefrontal cortex mediates a decrease in the release of striatal dopamine, while the decrease in cannabinoid receptor 1 (*Cnr1*) mediates the potentiation of brain damage.

Research examining sex differences in psychological dysfunction following TBI has indicated that estrogen and progesterone modulate the activity of GABA, dopamine, and serotonin systems, pointing to these pathways as potential targets for future investigation [19,119,120,121]. These findings are further supported by changes in calcitonin gene-related peptide (CGRP) levels, which influence the increase in Nrf2 expression and the reduction of psychological dysfunction observed in female rats [127], as well as alterations in CRF receptor expression [113,117]. Lastly, TBI is associated with increased expression of DNA methyltransferase (DNMT) and histone deacetylase 2 (HDAC2), along with decreased expression of growth arrest and DNA damage 45 (GADD45) [130,131]. Notably, while the BDNF IV promoter shows increased binding of HDAC2 and higher levels of histone H3 lysine 9 acetylation (H3K9ac) in both acute and chronic TBI, the BDNF IX promoter exhibits increased HDAC2 binding and decreased H3K9ac levels [130]. These epigenetic modifications result in reduced expression of BDNF IV and IX, which correlates with the development of anxiety.

Pre-clinical trends in therapeutic intervention development reflect the neurobiological mechanisms that have been assessed, targeting inflammation, oxidative stress, neurotransmission, sex hormones, and stress hormones (Figure 4, Table 3). Neurotransmission appears to be the most frequently targeted mechanism, focusing on the modulation of GABA, NMDAr, glutamate uptake, and the endocannabinoid system [92,95,99,105]. Interestingly, this largely follows the trends established by the treatment of injury-independent psychological dysfunction, which also appears to focus on neurotransmission as a therapeutic intervention. Beyond neurotransmission, however, interventions also target inflammation and/or oxidative stress, via the decrease in lipid peroxidation and inflammatory cytokines [72] or the modulation of glutathione expression [89], PKA/CREB [78], or Sepina3n [85]. Hormonal modulation has also shown a certain degree of promise, with the administration of progesterone decreasing anxiety-like behavior in rats [119], and the blockade of CRFR1 attenuating anxiety-related symptoms and HPA axis reactivity in mice [113].

### 2.3. Ischemic Stroke

#### 2.3.1. Clinical Presentation and Clinical Therapeutic Efficacy

Ischemic stroke affects over 75 million individuals worldwide annually, with its incidence steadily increasing [155]. It is characterized by a disruption in cerebral blood flow, leading to an initially reversible loss of function, which subsequently progresses to cerebral infarction [156]. This ischemic event triggers a cascade of pathological processes, starting with electrical dysfunction, followed by membrane disruption, calcium influx, excitotoxicity, reactive oxygen species (ROS) generation, and culminating in cell lysis and death. Psychological dysfunction is a frequent comorbidity following ischemic stroke. Meta-analyses show that between 39% and 52% of stroke patients develop one or more depressive episodes within the first five years post-stroke [157], and up to 30% develop anxiety within the first year [158]. The development of post-stroke depression (PSD) is influenced by multiple factors, including a history of depression and the severity and location of the stroke [159]. PSD is often associated with more severe depressive symptoms than generalized major depressive disorder, and the efficacy of treatments remains inconsistent and moderate [160]. Current pharmacological treatments for PSD include antidepressants, SSRIs, TCAs, CBT, and neuromodulation. However, the most effective treatment approach seems to involve a combination of pharmacological, psychosocial, and stroke-focused interventions, although these interventions tend to show higher rates of attrition due to the significant patient involvement required. Anxiety treatments following stroke typically involve SSRIs, TCAs, MAOIs, psychotherapy, acupuncture, exercise, and CBT. However, the greater emphasis on PSD research has limited the number of randomized controlled trials with placebo controls for anxiety, resulting in low-quality evidence with a high risk of bias regarding the efficacy of therapeutic interventions [26].

#### 2.3.2. Neurobiological Mechanisms and Pre-Clinical Therapeutic Targets

Preclinical studies of ischemic stroke have identified a range of mechanisms that contribute to the psychological dysfunction observed in the post-stroke period. These include apoptosis, neuroinflammation, oxidative stress, disruptions in neurotransmission, alterations in stress hormones, trophic factors, and DNA methylation (Table 1). Clinical research further suggests that synaptic plasticity and neuropeptides may also play a significant role in the development of psychological symptoms (Table 2). Given the extent of the primary injury, it is expected that mediators of this injury include pathways such as Akt1 signaling, the Nrf2 pathway, caspases 3, 8, and 9, HDAC3, and glial activation, all of which have been implicated in PSD [24,67,69,79,82]. Unique from TBI, dysfunction of the Akt/mTOR pathway has been observed to mediate anxiety following ischemic stroke [67], while increased expression of caspases 3, 8, and 9 in the hypothalamus, prefrontal cortex, and hippocampus mediate the presentation of PSD [69]. Separate from the effect of apoptosis, neuroinflammation, and oxidative stress, there is also a reduction in the m6A demethylase fat mass and obesity-associated protein (FTO) in the acute phase of ischemic stroke, accompanied by hypermethylation of N6-methyladenosine [129]. This epigenetic modification exerts significant downstream effects, as FTO regulates critical factors such as BDNF, a subunit of the NMDA receptor, myelocytomatosis oncogene, Jun, and oligodendrocyte transcription factor 2. FTO-mediated m6A demethylation is therefore essential for the regulation of neurogenesis, gliogenesis, axonal growth, synaptic plasticity, and stress responses. These changes are reflected in the altered expression of NMDA receptors, decreased serotonin levels, reduced CB1 receptor expression, increased CRH receptor expression, increased NGR, and disruptions in BDNF signaling within the hippocampus [82,100,103,106,114,115,116,118,125]. Clinically, elevated cytokine levels in saliva, cerebrospinal fluid (CSF), and serum have been associated with the onset of PSD, as have increased ferritin, cortisol, ApoE, substance P, and imbalances in neurotransmitter systems [139,140,142,143,144,145,146,147,148,149,152,153,161].

Much like in TBI, therapeutic targets in ischemic stroke include the overarching categories of neurotransmission, inflammation, oxidative stress, stress hormones, and trophic factors (Figure 4). Again, the most targeted mechanism is neurotransmission, with dopamine, glutamate, and serotonin presenting as promising interventional foci for PSD [91,94,97,103]. Oxidative stress and neuroinflammation are then the second-most targeted, with researchers focusing on HDAC, lipid peroxidation, and Nrf2, for a combined effect on both PSD and anxiety [79,82]. Lastly, BDNF and stress receptors have presented as potential targets for depression and anxiety, respectively [114,116,125].

### 2.4. Intracerebral Hemorrhage

#### 2.4.1. Clinical Presentation and Clinical Therapeutic Efficacy

ICH accounts for approximately 10 to 15% of all stroke types [162], with an estimated 3.4 million global cases reported in 2019, reflecting a substantial global health burden [163]. ICH is regarded as the most fatal form of stroke and is characterized by the formation of a hematoma within the brain parenchyma, with or without extension of blood into the ventricles. The damage results from both the spread of blood within the brain tissue and ischemia due to impaired cerebral perfusion. Consequently, 15 to 30% of patients experience depression and/or anxiety within the first year following ICH [164,165]. Treatment options include solution-focused approaches, SSRIs, TCAs, and anxiolytics, targeting behavioral modification, serotonin and norepinephrine uptake, and GABA [164,166,167,168]. While clinical trials have assessed the efficacy of these interventions, comprehensive evaluations have predominantly focused on SSRIs. Although SSRIs are effective in treating depression after ICH, their use is associated with an increased risk of ICH recurrence and poor neurological outcomes, thus limiting their application in certain patients [167,169]. 

#### 2.4.2. Neurobiological Mechanisms and Pre-Clinical Therapeutic Targets

Psychological dysfunction following ICH has predominantly been studied through the mechanisms of neuroinflammation, oxidative stress, and neurotransmission (Table 1), likely due to the lower incidence rate and the emphasis on primary injury rather than secondary complications. Neuroinflammatory processes, particularly glial activation, dysregulation of PTEN-induced putative kinase 1 (Pink1) and Nrf2 pathways, and cytokine release, have been strongly associated with the development of anxiety and depression after ICH [24,73,76,80,81,83,86]. In fact, Pink1 and Nrf2 have been independently identified as potential upstream mediators following ICH, their downregulation mediating the increased expression of monocyte chemoattractant protein-1, macrophage inflammatory protein, and proinflammatory macrophages, as well as decreased TrkB and BDNF expression. Concurrently, oxidative stress-induced damage is exacerbated by increased lipid peroxidation and a reduction in antioxidative enzymes such as glutathione (GSH), superoxide dismutase (SOD), and glutaminase, as again potentially mediated by Nrf2 [72,74,88]. Additionally, neurotransmission alterations are evident, with a reduction in dopaminergic neurons mediated by α-tubulin, impaired glutamate uptake, decreased Nrf2 activity, and lowered serotonin levels [74,76,81,101,162]. These changes are widespread throughout the brain.

The focus on neuroinflammation, oxidative stress, and neurotransmission is reflected in the therapeutics, which have been assessed pre-clinically for initial efficacy (Figure 4). These include hydrogen gas (H_2_) [73], Nrf2 [81,83], TRPV4 channel opening [74], and serotonin [101]. H_2_ administration decreases depressive symptomology in murine ICH via the increase in SOD, decrease in ROS, and downregulation of inflammatory cytokines, such as NF-κβ, IL-1β, and IL-6 [73]. The effect is significantly more apparent in female mice and is potentiated via the concurrent administration of estrogen in male mice, indicating a potential adjunctive target. Nrf2 induction via either pharmacological means or by enriched environment also resulted in the reduction of depressive symptomology in mice, presumably via the upregulation of BDNF and downregulation of inflammatory responses, ROS, and glutaminase activity [81,83]. Similarly, TRPV4 inhibition resulted in the reduction of astrocytic activation, lipid peroxidation, and total SOD, demonstrating a decrease in the anxiogenic effects of ICH [74]. Lastly, in terms of neurotransmission, inhibiting indoleamine 2, and 3-dioxygenase to prevent the downregulation of serotonin levels following ICH decreased stress presentation in mice [101].

### 2.5. Subarachnoid Hemorrhage

#### 2.5.1. Clinical Presentation and Clinical Therapeutic Efficacy

SAH represents a major global health challenge, with an estimated 8 million cases annually and an incidence rate projected to increase in parallel with the aging global population [170]. Meta-analytic studies have reported that approximately 28% and 32% of SAH patients experience depression and anxiety, respectively [171,172]. However, clinical randomized trials in SAH primarily focus on traditional outcomes such as aneurysm recurrence, retreatment, and mortality, rather than addressing psychological dysfunction [173]. As a result, there is limited clarity regarding the specific interventions for psychological dysfunction in this population and the efficacy of these treatments.

#### 2.5.2. Neurobiological Mechanisms and Pre-Clinical Therapeutic Targets

Preclinical investigations have examined psychological dysfunction in the context of apoptosis and neuroinflammation, in conjunction with clinical assessments of stress hormone and neuropeptide levels (Table 1 and Table 2). The mediation of anxiety, compulsive-like behaviors, and depressive symptoms through apoptosis is driven by ERK and P2Y14 receptor signaling, particularly within the hypothalamus and cortex, respectively [65,68,69]. ERK signaling mediated by TNFα is actually necessary for hypothalamic apoptosis and the production of anxiety [68]. Depression has also been linked to impaired glutamate uptake in the striatum [97]. Clinically, alterations in neuropeptide and stress hormone levels are observed globally, with these changes manifesting as measurable variations in urine, serum, and cerebrospinal fluid, correlating with psychological dysfunction [141,151]. These include an increase in admission CGRP within the CSF correlating to poorer long-term psychological outcomes [151] and the increase in early cortisol associating with late-stage depression [141]. Reflecting the decreased assessment of mechanisms of psychological dysfunction following SAH, only apoptosis, and neurotransmission have been targeted as potential therapeutic interventional foci in pre-clinical research (Figure 4). To be exact, acute administration of hydrogen gas inhibited the p38 MAPK pathway to prevent apoptosis, and the resultant anxiety and depression [66], while the acute modulation of glutamate uptake via HDAC2 has been found to decrease depression [97].

### 2.6. Vascular Cognitive Impairment and Dementia

#### 2.6.1. Clinical Presentation and Clinical Therapeutic Efficacy

VCID accounts for approximately 20–40% of all dementia diagnoses and is primarily caused by cardiovascular disease, which is associated with vascular risk factors leading to chronic cerebral hypoperfusion (CCH) [174]. CCH is a common pathophysiological condition, often resulting from the chronic reduction in cerebral blood flow, leaving the brain in a state of prolonged hypoxic ischemia. This condition contributes to progressive neurological and cognitive dysfunction [175]. In addition to cardiovascular disease, CCH can also develop as a long-term consequence of acute brain injuries. In 2018, the global prevalence of VCID was estimated at 50 million individuals, with projections suggesting a threefold increase in the next 25 years, particularly in high-income countries. Large-scale studies of VCID report that at least 80% of patients exhibit some form of neuropsychiatric symptom [176,177]. While many VCID treatments focus on addressing cognitive impairment, rather than neuropsychiatric symptoms, certain interventions, including NMDA antagonists and SSRIs, have been evaluated for their efficacy in mitigating psychological dysfunction, although their effectiveness in VCID remains limited [178,179].

#### 2.6.2. Neurobiological Mechanisms and Pre-Clinical Therapeutic Targets

CCH is characterized by a gradual, chronic development of injury. Although it might be expected that unique mechanisms would contribute to psychological dysfunction in CCH, this is not entirely the case. Apart from amyloid and cerebral insulin metabolism, CCH shares several mechanisms of psychological dysfunction with acute brain injuries (Table 1 and Table 2). However, in CCH, the primary focus shifts from apoptosis, neuroinflammation, and oxidative stress to alterations in neurotransmission and synaptic plasticity, particularly within the hippocampus. Multiple aspects of these processes are implicated in the development of anxiety and depression in CCH, including aberrant GABA and glutamate signaling, reduced BDNF, impaired synaptic potentiation, changes in synaptophysin, serotonin, and HCN1, as well as reduced dopamine and KCNQ3 activity [21,22,77,91,102,104,107,108,109,111]. Uniquely, the decrease in CNPase and synaptophysin in the white matter mediate an impairment of synaptic potentiation and in anxiety and depression [111], alongside the upregulation of HCN1 and downregulation of KCNQ3 in the amygdala [104]. While neuroinflammation and oxidative stress are not overlooked in the study of psychological dysfunction in CCH, only specific factors are typically assessed, including phosphorylation of TrkB, ERK, CREB, antioxidant expression, and lipid peroxidation [87], which reflect previously discussed patterns. Additionally, the upregulation of the angiotensin-converting enzyme/angiotensin II/angiotensin II type 1 receptor (ACE/Ang II/AT1R) axis in the hippocampus, as opposed to the neuroprotective angiotensin-converting enzyme 2/angiotensin-(1–7) (ACE2/Ang-(1–7)) axis results in decreased cerebral blood flow and increased oxidative stress and neuroinflammation, which contributes to the manifestation of anxiety [128].

Notably, CCH is the only model in which a link between psychological dysfunction and increased tau metabolism or decreased insulin metabolism has been identified, potentially reflecting the chronic nature of VCID [122,123,124]. Amyloid beta peptide (Aβ) deposition induces neuronal damage and has been linked to behavioral and anxiety disorders in CCH. Specifically, β-amyloid precursor protein cleavage enzyme 1 (BACE1), which functions as the β-secretase responsible for processing amyloid precursor protein, is upregulated after cerebral ischemia [123]. This upregulation of BACE1 is associated with increased Aβ levels in the hippocampus [122]. Abnormal hyperphosphorylation of Tau is mediated by reduced levels of tau O-GlcNAcylation, a post-translational modification involving β-N-acetylglucosamine (GlcNAc) [124]. Alongside GlcNAc dysfunction, the downregulation of the insulin signaling pathway components—such as the insulin receptor, IGF-1 receptor, insulin receptor substrate-1, phosphatidylinositide 3-kinases, 3-phosphoinositide-dependent protein kinase-1, protein kinase B, and glycogen synthase kinase-3—has been associated with anxiety-like behavior following cerebral ischemia [124].

Reflecting the differing investigative foci of CCH, therapeutic interventional research has largely focused on the modulation of synaptic plasticity, though neurotransmission, oxidative stress, neuropeptides, and neuroinflammation have also been targets (Figure 4). In regards to synaptic plasticity, specific neurobiological mechanistic targets for anxiety and depression include transforming growth factor β, BDNF, glycogen synthase kinase-3β, synaptophysin, and lipid peroxidation [77,107,108,109,111]. Neurotransmitter targets include GABA receptors in the amygdala and hippocampus and serotonin within the hippocampus [21,102], while the modulation of SOD and glutathione decreases anxiety-like symptoms [87]. Lastly, modulation of the angiotensin system decreases oxidative stress and neuroinflammation, resulting in the ablation of depressive symptomology [128].

## 3. Sexually Dimorphic Responses Reveal Potential Targets for Therapeutic Intervention in Psychological Dysfunction Within Injured Brains

Sex serves as an independent modulator of psychological dysfunction symptomatology and treatment efficacy, regardless of the presence of cerebral injury. Epidemiological studies indicate that absent of injury, the female population is more likely to develop depression, anxiety, or PTSD than males [34,180,181,182]. Similarly, older females show an increased incidence of depression and anxiety following ischemic stroke and SAH, with a comparable trend noted in pediatric TBI [18,20,23,183]. However, in the adult military population, mTBI is linked to a higher prevalence of anxiety in males than females [184]. The divergence in sex differences between pediatric and adult mTBI may arise during adolescence, as females show reduced severity of psychological dysfunction following mTBI during mid-puberty [185]. In pre-clinical observations, female rodents exhibit lower depression and anxiety levels in blast injury and severe TBI [19,117,119,120,127], as well as enhanced responses to therapeutic intervention in ICH [73]. The factors underlying these observed differences are not limited to sex hormones and may represent novel targets for therapeutic intervention.

Sexually dimorphic presentations of psychological dysfunction have been linked to multiple neurobiological mechanisms. In non-injured contexts, estrogen appears to reduce the impact of stress on cognitive function in females [33], yet fluctuations in estrogen levels, particularly during menopause, are associated with an increased risk of developing depression [36]. Under healthy conditions, females also exhibit heightened activation of the locus coeruleus and greater sensitivity to CRF within the region [33,34,35]. While this may increase susceptibility to hyperarousal in females, it also presents CRF as a potential target for therapeutic intervention. Pre-clinical examinations within blast injury TBI support this supposition, with the observation that variations in CRFR2 between males and females lead to reduced anxiety levels [117]. While expression within the locus coeruleus itself was not measured, it was observed that female animals exhibit decreased CRFR2 in the dentate hilus of the hippocampus following blast TBI, while males exhibit increased expression in the ventral hippocampus and decreased expression in the amygdala. In addition to stress hormones, neuropeptides may be potential targets of therapeutic intervention. In a severe TBI model, female rodents exhibit both decreased anxiety and decreased depression; this is mediated by the action of CGRP [127]. The increased expression of CGRP not only reduces oxidative stress and vascular damage in female brains, but females also exhibit a more pronounced response to CGRP inhibition compared to males. This suggests that female animals have developed adaptive mechanisms reliant on elevated CGRP expression and sensitivity, which, if replicable, could be exploited as a broad therapeutic strategy.

Sex hormones themselves also present an attractive potential therapeutic target. Clinically, the higher female anxiety observed in pediatric TBI is linked to increased extracellular diffusivity, which is not observed in females with adult hormone levels [150,186]. In males, however, decreased testosterone levels in military males following TBI are associated with the development of anxiety [184,187]. Preclinically, estradiol and progesterone have been shown to influence anxiety in ICH and TBI by modulating neurotransmission, oxidative stress, and inflammation [73,119,188]. Estradiol impacts GABA_A_ receptor levels, serotonin signaling, mitochondrial function, and antioxidant expression [119,188], while in ICH estrogen modulates astrocyte-mediated inflammation, which may influence the development of depression [73]. While there have been mixed results in mortality and neurological outcomes in clinical trials administering estrogen and progesterone in TBI [127], this may be mediated by other factors, namely severity of brain injury, which ought to be taken into account alongside hormone levels. Meta-analyses of preclinical studies indicate that lower TBI severity is associated with poorer outcomes in females, while moderate-to-severe TBI often correlates with improved outcomes in females [20]. This suggests that injury severity may influence the extent of sexual dimorphism and highlights the significant role of injury severity and etiology in determining the response to therapeutic interventions. Additionally, these findings challenge the assertion that the mechanisms of psychological dysfunction are consistent across various types of brain injury [26], despite differences in etiology. Given the observed sexual dimorphism in response to TBI severity, it is crucial to consider both the type and severity of injury when selecting and evaluating therapeutic interventions for psychological dysfunction following brain injury.

## 4. Future Directions for Enhancing Therapeutic Efficacy in Injury-Dependent Psychological Dysfunction

### 4.1. Targeting Upstream Mechanisms as a Comprehensive Therapeutic Strategy

Throughout this review, it has become evident that while current therapeutic approaches for psychological dysfunction are diverse and effective in injury-independent contexts, they are not fully effective in addressing injury-dependent psychological dysfunction [58,189,190,191,192,193]. CBT, for example, is a first-line treatment in non-injured individuals with anxiety and depression [194], but its reliance on active patient engagement diminishes its utility in injured contexts [189,190]. Neuromodulation, which concurrently targets multiple neurobiological mechanisms, shows promise but remains underutilized in the clinical treatment of psychological dysfunction following brain injury [58,189,190]. Vagus nerve stimulation, trigeminal nerve stimulation, and transcranial magnetic stimulation, among others, have shown positive effects in treating anxiety and depression in non-injured individuals [58,195,196,197], however, only transcranial magnetic stimulation has been studied more extensively in injured populations [198,199]. Consequently, in the context of injury-dependent psychological dysfunction, pharmacological interventions remain the primary treatment approach. Many of these treatments, however, exhibit reduced therapeutic efficacy, potentially due to their focus on downstream factors, like serotonin dysregulation, which does not address the factors driving the dysregulation. Targeting multiple underlying mechanisms pharmacologically, however, presents a challenge, as pharmaceutical cocktails can result in potentially harmful drug–drug interactions [200,201]. The prevalence of side effects with such combinations limits their clinical application and increases the necessary testing and validation. Therefore, the identification of a comprehensive, multi-target approach capable of simultaneously modulating the various factors involved in psychological dysfunction may represent a promising avenue for future research.

One approach to targeting the multiple, divergent pathophysiological cascades that contribute to psychological dysfunction may involve focusing on upstream mediators (Figure 5). Nrf2, for instance, has emerged as a promising target in depression research and in the treatment of brain injury [202,203,204,205,206]. As an upstream regulator, Nrf2 modulates key factors involved in the development of psychological dysfunction (Table 1), including neurotransmitters such as serotonin and dopamine, which are commonly targeted by pharmacological therapies, as well as processes like apoptosis, neuroinflammation, and oxidative stress. By concurrently modulating factors involved in both injury progression and the development of psychological dysfunction, Nrf2 emerges as a promising therapeutic target for injury-associated psychological dysfunction. This has already been assessed as a target for PSD, with controlled regulation of Nrf2 activation, via intravenous application of CDDO-Im, resulting in decreased depressive symptomology in a murine PSD model, as well as an increase in autophagic factors [82]. Further, in ICH, Nrf2 induction, via either an enriched environment or the application of TP-500, results in the reduction of depressive symptomology in mice [81,83]. This was mediated via an effect on BDNF and the downregulation of inflammatory responses, ROS, and glutaminase, illustrating the potential of Nrf2 to modulate neurotransmission, neuroinflammation, and oxidative stress. This indicates Nrf2′s promise as an interventional target in different forms of brain injury, addressing multiple factors affecting the presentation of psychological dysfunction.

Nrf2 is not the only upstream mediator that may be a promising target for the treatment of psychological dysfunction. Neuropeptides are multi-focal molecules packaged and co-released with neurotransmitters, the functional implications of which have been studied extensively in numerous reviews [32,207,208,209,210]. The neuropeptides connected to psychological dysfunction in non-injured contexts are multitudinous, including CCK, NPY, galanin, oxytocin, and vasopressin [32]. When examining injured contexts, the list increases to include CGRP, angiotensin II, and substance P [127,128,151], while both CCK-8 and NPY are also implicated in PSD [147,152,153]. Together, these molecules have the capacity to mediate changes in oxidative stress, inflammation, the stress axis, vasoreactivity, apoptosis, neurotransmission, and synaptic plasticity [211] (Figure 6). CGRP, for example, can modulate ERK, PI3K/AkT, FoxO3a, CREB, MAPK, BDNF, glial activation, and cytokine release [212]. This is in addition to the activation of Nrf2, which angiotensin II, NPY, and substance P also mediate [127,213,214]. Targeting angiotensin II and the angiotensin system in pre-clinical CCH has already demonstrated an ablation of depressive symptomology, as mediated by a decrease in inflammation and oxidative stress [128]. However, the relative lack of investigation into the therapeutic effect of these molecules belies their apparent promise and indicates the necessity for investigation in the context of injury-induced psychological dysfunction.

As epitomized in the usage of enriched environments to enhance Nrf2 activation and reduce depressive symptomology in ICH, the integration of non-pharmacological therapeutics may offer substantial benefits (Figure 7). These approaches influence multiple neurobiological mechanisms involved in the development of psychological dysfunction following brain injury, including the concurrent modulation of neuroinflammation, neurotransmission, apoptosis, oxidative stress, neuropeptides, and synaptic plasticity [58,215,216], all of which independently represent attractive targets for treatment of psychological dysfunction following injury. Both physical exercise and various forms of neuromodulation have also been shown to induce these effects and may thus be relevant in managing psychological dysfunction. For instance, trigeminal nerve stimulation modulates neuropeptides, including the aforementioned CGRP, NPY, and substance P, neuroinflammation, via the modulation of microglia, leukocytes, and Th1 cells, and neurotransmission, mediated by changes in NMDAr and glutamate and GABA signaling [58]. Currently, it is already under clinical investigation for the treatment of non-traumatic depression, social anxiety disorder, PTSD, and loss of consciousness following brain injury, and is being explored preclinically as a potential treatment for SAH, TBI, and ischemic stroke. Physical exercise, on the other hand, is known to have positive effects on mental health in non-injurious conditions, mediating a 43.2% decrease in self-reported mental health dysfunction among 1.2 million individuals [217]. Not only does it mediate endorphin release, but it also modulates the HPA axis, neurotransmitter activity, cytokines, vagal tone, toll-like receptors, and mitochondria, as well as mirroring effects of CBT via providing a sensation of mastery and a distraction from negative thoughts and ruminations [218]. These examples thus exemplify a theoretical multi-mechanistic strategy for addressing psychological dysfunction following brain injury and illustrate non-pharmacological interventions as a promising area for further experimental investigation, alongside upstream modulators.

### 4.2. Tailoring Therapeutic Interventions to Address Psychological Dysfunction Based on the Specific Type of Brain Injury

This review reveals that psychological dysfunction following brain injury is often treated similarly to injury-independent psychological dysfunction. While this approach leverages established knowledge, it may not fully address the unique complexities of injury-related psychological dysfunction. For instance, in TBI, both injury severity and age at the time of injury can influence psychological outcomes, with sex-specific differences observed [20,185]. This suggests a pathophysiological cascade influenced by injury severity, which interacts with sex-related factors to differentially affect symptomatology. Therefore, it is unlikely that mechanisms in injury-dependent contexts are identical to those in injury-independent psychological dysfunction [26]. Additionally, the assertion that psychological dysfunction mechanisms are similar across various brain injury types is questionable. For example, NMDAr dysfunction plays a role in psychological dysfunction in both SAH and TBI [93,94,95,96,97,99,105]. However, due to differences in injury development, a single NMDAr-targeting intervention may not be effective across both conditions. In SAH and ICH, NMDAr dysfunction is directly mediated by thrombin, which potentiates NMDAr function, worsening glutamate-induced cell death [219,220]. Thrombin levels remain elevated for up to nine days after SAH, indicating prolonged NMDAr potentiation [221]. In contrast, TBI exhibits distinct temporal patterns, with NMDAr expression initially increasing in the hyper-acute phase but decreasing in the chronic phase [93,94,95,96,99,105]. Thus, therapies targeting NMDAr activity may be beneficial for SAH/ICH but harmful in TBI. This highlights the importance of considering injury-specific contexts, as preclinical studies on SAH have primarily focused on neurotransmission and apoptosis [65,66,69,70,97], while clinical conditions also involve neuroinflammation, stress hormones, and neuropeptides [141,151]. This suggests that the assessment of underlying mechanisms of psychological dysfunction is being influenced more by observations from other types of brain injuries, rather than being specifically tailored to the injury in question. Given the diverse etiologies and disease progression patterns [40,44,46,49,51,54](Figure 2), it is unlikely that the brain regions and mechanisms affected are the same across all types of brain injury. This raises concerns about the current research approach, which often evaluates similar mechanisms across different brain injury models (Table 1). As an initial step, we recommend concurrently examining the mechanisms and presentations of psychological dysfunction in multiple models to identify potential discrepancies, which could inform the development of more targeted therapeutic interventions. Additionally, we advocate for considering psychological dysfunction in various brain injuries as separate entities to prevent conflating results between models.

A further complicating factor is the observation that in pre-clinical research, current therapeutic interventions do not appear to be tailored to the state of disease progression (Table 1). Instead, therapies applied during the chronic phase of disease progression target the same mechanisms as those used in the acute phase, focusing on oxidative stress, neuroinflammation, neurotransmission, and DNA methylation (Table 1 and Table 3, Figure 4). A review of the relevant studies reveals that interventions were selected based on their demonstrated efficacy in the acute phase of the disease [72,82,103,113,130]. There was no indication that these interventions were specifically designed to address the unique pathophysiological cascades associated with the chronic phase, implying that these approaches treat acute and chronic stages in similar contexts. However, as illustrated by the pattern of NMDAr changes in TBI [93,94,95,96,99,105], and the pathological development due to blood presence after SAH and ICH [221], brain injury leads to ongoing pathological changes over time, which are reflected in evolving patterns of psychological dysfunction. Fundamentally, therapeutic interventions following brain injury ought to be tailored to the stage of disease progression, either by adjusting interventions over time as the injury or recovery progresses or by using a single intervention that can be modulated to suit temporal changes in the disease state, such as neuromodulation and physical exercise. In addition to their ability to address multiple underlying mechanisms and their proven safety in clinical settings [58,215,216], neuromodulation offers the unique advantage of modulating treatment intensity, making it adaptable to different injury severities and stages of recovery. This flexibility provides a potential avenue for managing psychological dysfunction in individuals recovering from brain injuries.

### 4.3. The Implementation of an Integrative Theoretical Framework to Address the Limitations of Current Approaches That Contribute to Diminished Therapeutic Efficacy

The development and treatment of injury-independent psychological dysfunction is inherently complex. Previous studies on psychological dysfunction and its treatments have primarily focused on symptomatology and clinical presentation, often overlooking the underlying etiological factors, which led to the application of imprecise or ineffective therapies [222]. In response, integrated approaches have emerged that not only address symptoms but also examine environmental and neurobiological factors, alongside personal history, to construct a comprehensive model of disease development [1,222,223,224]. These frameworks have enhanced the understanding of causal factors, allowing for targeted interventions that address the most relevant elements. For example, the reciprocal interaction model for obsessive–compulsive disorder considers neuropsychological deficits, cognitive-behavioral processes, and the source of the compulsions, enabling the identification of disease subtypes, triggers, and their underlying causes, while facilitating the construction of personalized treatment plans [225]. Similar integrative frameworks for depression have incorporated environmental factors, regional “kindling” by induced stress or trauma, HPA axis activity, stress levels, and personal vulnerability [1,223]. These models not only identify neurobiological targets for treatment (e.g., serotonin or dopamine) but also incorporate non-pharmacological interventions, such as CBT, to address triggers or environmental factors, leading to more effective interventions than those based on symptom-driven treatments alone [222].

Following brain injury, the factors influencing the development and presentation of psychological dysfunction become more intricate, with the added effect of multiple pathophysiological cascades resulting from tissue damage. For instance, the decreased efficacy of dopamine and serotonin in the context of injury in clinical settings [26,134,136,137] may stem from the fact that they are not targeting the underlying factors, such as oxidative stress, neuroinflammation, apoptosis, or dysregulation of hormones and peptides (Table 1), that contribute to the altered levels of dopamine and serotonin. Consequently, simply administering SSRIs, benzodiazepines, or MAOIs potentially allows upstream dysregulation to persist and undermines therapeutic efficacy. However, due to understandable experimental limitations, current preclinical research often fails to adequately address the interacting causal factors (Table 1). Thus, the first step is to identify the most relevant factors to target, akin to strategies in injury-independent contexts [222]. By synthesizing current knowledge into an integrative theoretical framework, it is possible to inform therapeutic interventions and guide future research. This review itself reveals two key observations: (1) there are notable differences between injury-independent and injury-dependent psychological dysfunction, indicating that it is inappropriate to assume the same mechanisms are predominant in both contexts and (2) targeting factors directly associated with psychological dysfunction (e.g., serotonin and dopamine) is insufficient for addressing injury-dependent psychological dysfunction. These insights can directly inform future research, and a more comprehensive theoretical framework will likely raise additional areas for intervention.

In the vein of approaching injury-dependent psychological dysfunction in an integrated manner, it is crucial to: (1) adopt more clinically relevant models, which at least partially integrate the environmental factors and individual susceptibility (Table 1), and (2) examine psychological dysfunction in a manner encompassing the effects of multiple brain regions (Table 1). Currently, preclinical models typically isolate brain injury as the sole stressor [1,222,223,224]. While it is understandable to initially evaluate therapeutic interventions with brain injury alone as the stressor, the absence of additional stressors in animal models represents a limitation that can be addressed via the inclusion of a chronic stressor post-injury, such as chronic unpredictable mild stress [77], or the usage of animals subjected to maternal deprivation prior to injury [226,227]. An additional potential target for future research is the examination of different brain regions in injury-dependent psychological dysfunction. In brain injury, the location of the initial insult does not necessarily determine where secondary damage will occur, with primary damage to the cortex resulting in subsequent damage in anatomically distant regions, such as the amygdala or thalamus, in TBI [127]. Further, due to brain connectivity, damage in one region may lead to effects in other regions [49]. Therefore, limiting the examination of mechanisms to specific sub-regions may fail to capture critical interactional effects and cross-regional influences. As such, in TBI, research ought to expand beyond the cortex, hippocampus, and amygdala, which currently represent the majority of research in TBI-dependent psychological dysfunction (Table 1). Similarly, in VCID, the focus on the hippocampus, due to its role in cognition [178], largely neglects other brain sub-regions involved in behavior and emotional regulation [178,179]. While examining all brain regions at once is impractical, shifting the focus of psychological dysfunction research away from regions most associated with overt pathophysiology could facilitate the identification of more contextually appropriate interventions.

## 5. Conclusions

This review provides a comprehensive evaluation of the neurobiological mechanisms and therapeutic strategies related to neuropsychological dysfunction, with a particular emphasis on psychological disturbances arising from brain injuries. It highlights promising therapeutic approaches for addressing injury-dependent psychological dysfunction, identifying Nrf2, neuropeptides, and nonpharmacological therapies as potential multi-mechanistic interventions that act as upstream mediators. These interventions target the complex interactions of various factors contributing to psychological dysfunction in the context of brain injury. Additionally, the review underscores the importance of sexually dimorphic factors, suggesting them as key areas for further investigation and advocating for a deeper exploration of sex-specific patterns to identify additional contributors to psychological dysfunction. The review also identifies significant gaps in current research, particularly the limited consideration of the intricate interactions among causal factors, environmental influences, and individual susceptibilities. This highlights the need for an integrative framework to guide the identification of effective therapeutic targets and promote the use of clinically relevant models. Furthermore, it points to an over-reliance on findings from injury-independent contexts, the assumption that neurobiological mechanisms are consistent across different types of brain injury, and the prevalent focus on acute-phase therapeutic interventions in preclinical research. This approach neglects the distinct pathophysiological cascades associated with various forms and stages of brain injury and overlooks how factors such as the presence or absence of hemorrhage in acute brain injuries can influence the underlying mechanisms of psychological dysfunction. Moreover, it fails to address the chronic phase, during which psychological dysfunction typically emerges. To address these challenges, the review recommends prioritizing research on modulators of psychological dysfunction that reflect injury-specific deviations from non-injury patterns, considering factors such as injury severity and type. It also advocates for tailoring therapeutic strategies according to the stage of injury progression and the specific characteristics of the injury to optimize outcomes. In conclusion, the review emphasizes the need for multi-mechanistic, context-specific approaches to improve therapeutic outcomes and address existing gaps in the understanding of the neurobiological mechanisms of psychological dysfunction across different brain injuries.

## Figures and Tables

**Figure 1 cells-14-00074-f001:**
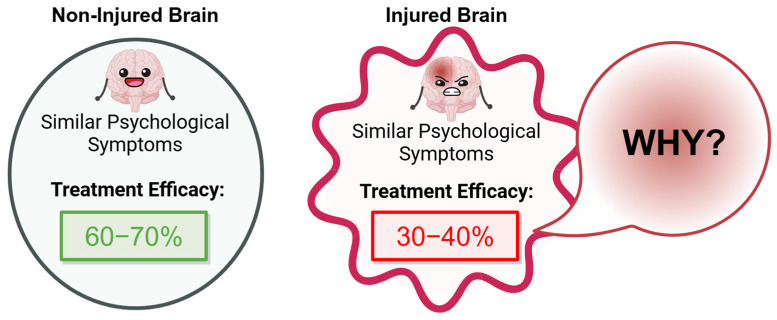
Therapeutic efficacy of psychological dysfunction in non-injured versus injured populations. While symptomatology is comparable, the therapeutic efficacy for psychological dysfunction is notably diminished in brain-injured populations compared to non-injured individuals. (Created with Biorender.com).

**Figure 2 cells-14-00074-f002:**
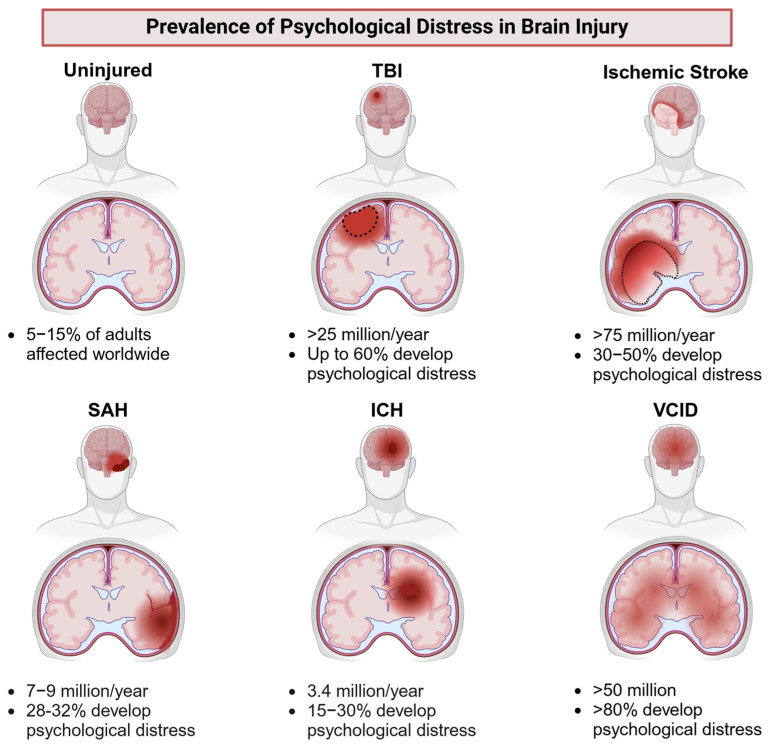
Different types of brain injury are associated with varying prevalences of psychological dysfunction. Psychological dysfunction is more prevalent in populations with brain injury, with each injury type exhibiting distinct rates and prevalences of psychological symptoms. Notably, vascular cognitive impairment and dementia (VCID), despite showing minimal overt damage (highlighted in red), exhibit the highest prevalence of psychological dysfunction. (Created with Biorender.com) (ICH: intracerebral hemorrhage; SAH: subarachnoid hemorrhage; TBI: traumatic brain injury).

**Figure 3 cells-14-00074-f003:**
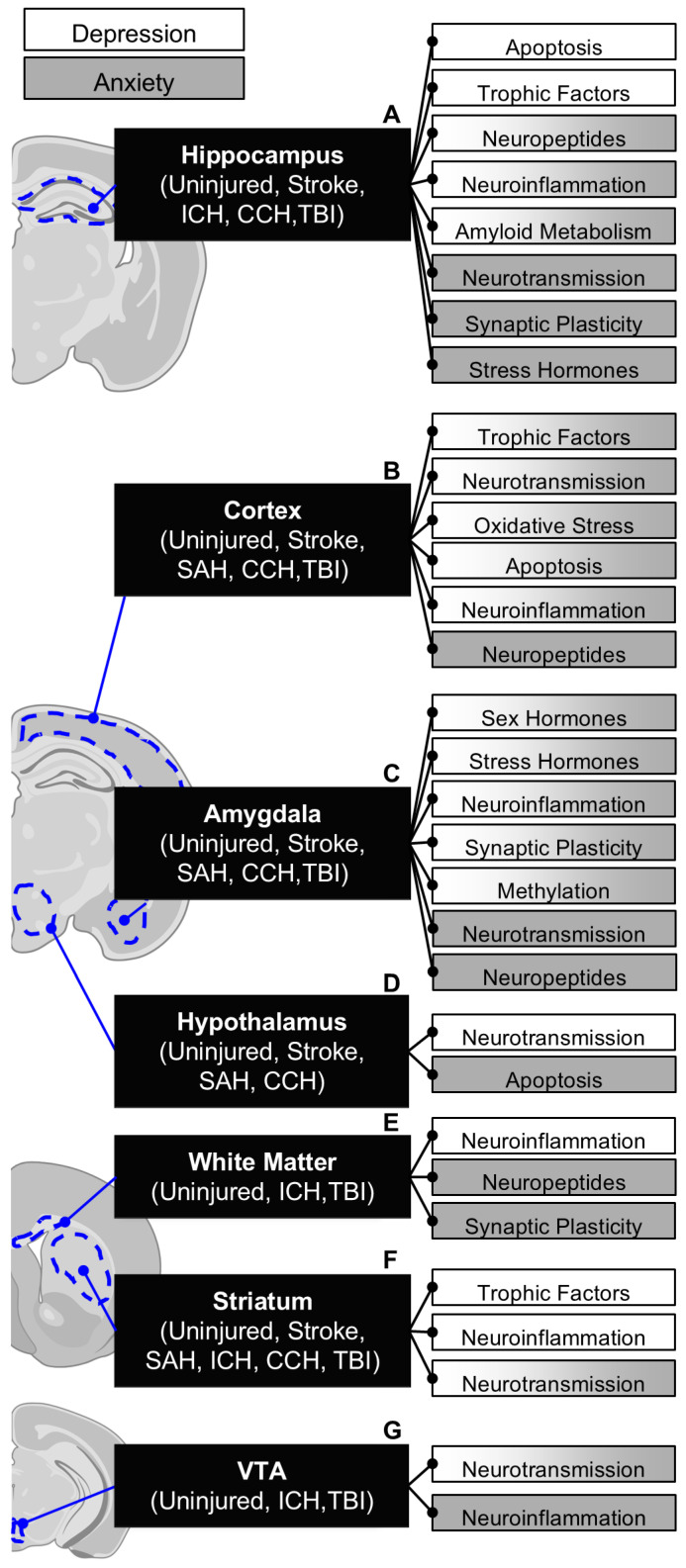
Psychological dysfunction is associated with alterations in specific brain regions responsible for emotional regulation, with each region contributing to distinct pathogenic cascades that collectively intensify psychological dysfunction. An analysis of studies on psychological dysfunction suggests that the brain sub-regions examined are not dependent on the location, severity, or presence of the primary injury. The primary regions investigated in relation to psychological dysfunction following brain injury include (**A**) the hippocampus, (**B**) the cortex, (**C**) the amygdala, (**D**) the hypothalamus, (**E**) white matter, (**F**) the striatum, and (**G**) the ventral tegmental area. Notably, the hippocampus, cortex, and amygdala are the main regions studied, while other areas involved in psychological regulation remain comparatively underexplored. (White boxes = depression, gray boxes = anxiety, gradient boxes = anxiety and depression) (Created with Biorender.com) (CCH: chronic cerebral hypoperfusion; ICH: intracerebral hemorrhage; SAH: subarachnoid hemorrhage; TBI: traumatic brain injury).

**Figure 4 cells-14-00074-f004:**
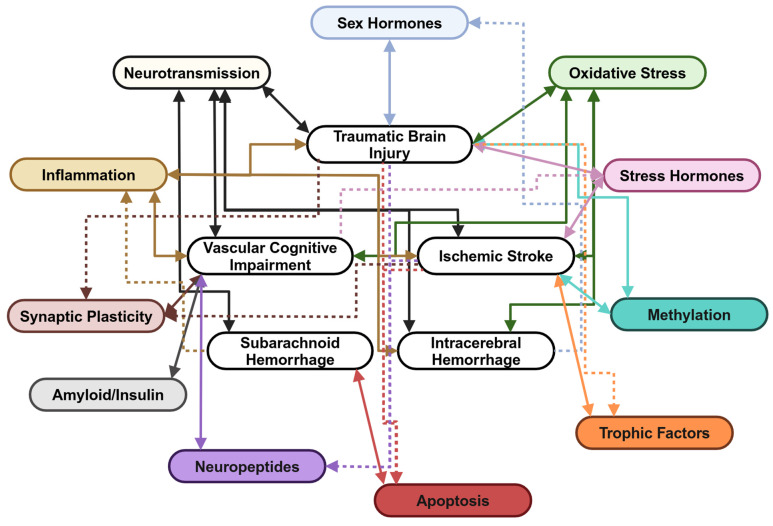
Therapeutic interventions targeting neurobiological mechanisms underlying psychological dysfunction in injury-related contexts. Eleven overarching categories of neurobiological mechanisms have been identified as contributing to psychological dysfunction in the context of brain injury. While there is considerable overlap across different types of brain injury, not all mechanisms have been evaluated in every form of brain injury. Additionally, among the assessed mechanisms, only a subset has been specifically targeted with the aim of influencing psychological function (solid, double-headed arrows = identified mechanisms that have been targeted, others have not been targeted at all for therapeutic intervention (dashed, single-headed arrows (see: Table 3, greyed out rows)).

**Figure 5 cells-14-00074-f005:**
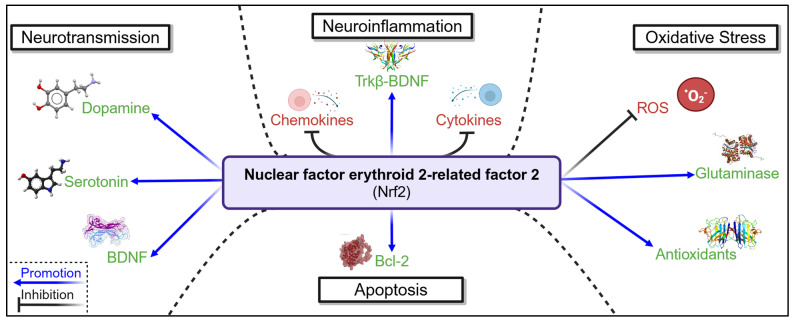
Nrf2 emerges as a promising upstream target for therapeutic intervention in psychological dysfunction associated with brain injury. To effectively address the multiple underlying mechanisms contributing to psychological dysfunction following brain injury, targeting upstream pathways may offer a promising direction for future research. Given its ability to simultaneously modulate neurotransmission, neuroinflammation, oxidative stress, and apoptosis, Nrf2 stands out as a potential target for therapeutic intervention. (Created with Biorender.com).

**Figure 6 cells-14-00074-f006:**
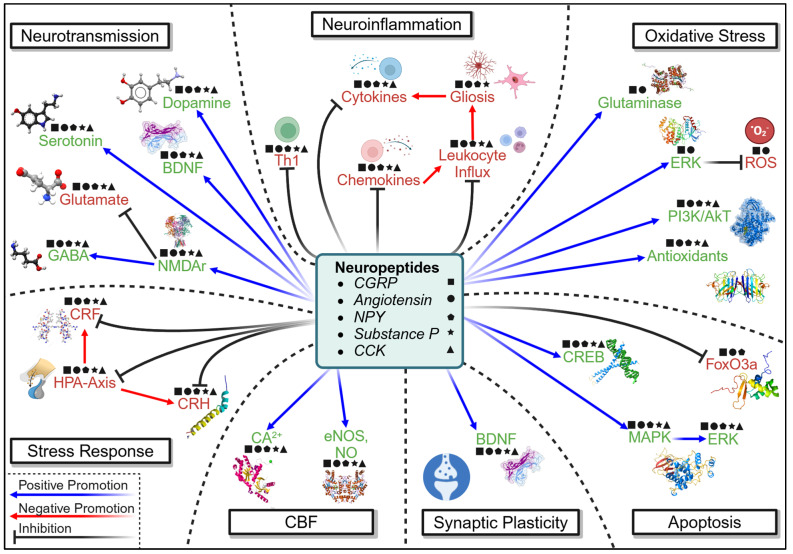
The potential of neuropeptides as promising upstream targets for therapeutic intervention in injury-dependent psychological dysfunction. Neuropeptides function as multifaceted molecules that regulate several processes within the nervous system, including neurotransmission, neuroinflammation, oxidative stress, apoptosis, synaptic plasticity, cerebral blood flow (CBF), and stress responses. These pathways are integral to the development of injury-dependent psychological dysfunction, highlighting the significant potential of neuropeptides as a target for therapeutic intervention. (Each mechanism is annotated with the corresponding neuropeptides that influence it, represented by specific icons: CGRP (square), angiotensin (circle), NPY (pentagon), substance P (star), and CCK (triangle)). (Created with Biorender.com).

**Figure 7 cells-14-00074-f007:**
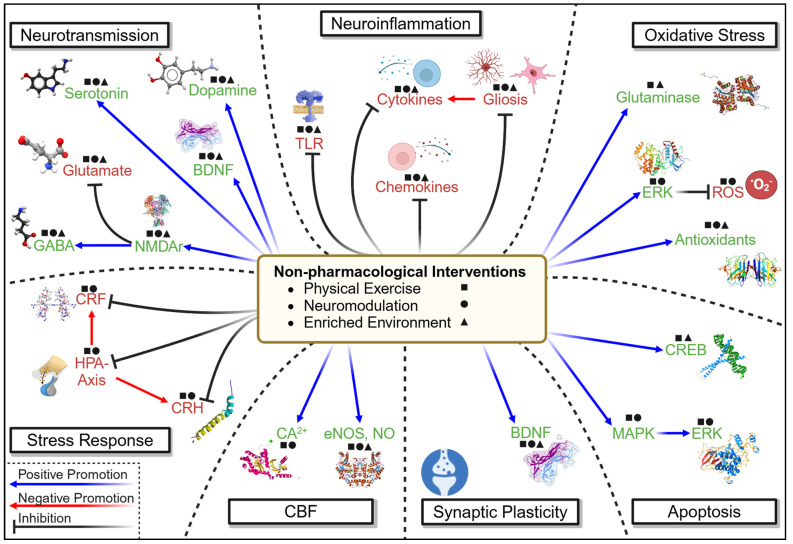
Non-pharmacological interventions as potential therapeutic strategies for alleviating psychological dysfunction following brain injury. Non-pharmacological interventions are inherently multi-mechanistic, enabling them to influence a broad array of factors that contribute to the onset of psychological dysfunction following brain injury. These factors include neurotransmission, neuroinflammation, oxidative stress, apoptosis, synaptic plasticity, cerebral blood flow (CBF), and stress responses, thereby targeting a significant portion of the pathways involved in the development of psychological distress. (Each mechanism is annotated with the interventions known to influence it, represented by specific icons: physical exercise (square), neuromodulation (circle), and an enriched environment (triangle)). (Created with Biorender.com).

**Table 2 cells-14-00074-t002:** Clinical studies investigating the neurobiological mechanisms underlying anxiety and depression in brains impacted by injury. (↑ = increase; ↓ = decrease; → = leads to).

Mechanism	Trends	Sex-Specificity	Global or Focal	Disease/Model	Ref.
Neuroinflammation	↓IL-6, ↑α-amylase → Depression	N/A	α-amylase: Saliva IL-6: Serum	Stroke	[139]
	↑IL-1β, ↑TNF-α → Depression	N/A	Serum	Stroke	[140]
	Elevated early CSF and sera IL-6 levels associated with fatigue	N/A	CSF/serum	SAH	[141]
Oxidative Stress	Ferritin → Depression	N/A	Serum	Stroke	[142]
Stress Hormones	↑early urinary free cortisol → depression, fatigue, and ↓ quality of life ↑serum cortisol, serum CRH → depression	N/A	Urine Serum	SAH Stroke	[141,142]
Synaptic Plasticity	↑Serum ApoE, ↓monocyte ApoE → Depression	N/A	Serum	Stroke	[143]
Neurotransmission	↓HTR3D, ↓NEUROG3 → Depression	N/A	Global	Stroke	[144]
	↑5-HTTLPr → Depression	N/A	Global	Stroke	[145]
	p11/tPA/BDNF → Depression	N/A	Global	Stroke	[146]
	↓Serotonin → Depression	N/A	Serum	Stroke	[147]
	↑Glutamate → Depression	N/A	Plasma	Stroke	[148]
	↓Glutamate → Depression	N/A	Plasma	Stroke	[149]
	↑Diffusivity → Anxiety and Depression	Pediatric females are more likely to develop ↑diffusivity	Amygdala	TBI	[150]
Neuropeptides	↑CGRP concentration → poorer psychological outcomes	N/A	CSF	SAH	[151]
	↓NPY → Depression	N/A	Plasma, serum	Stroke	[152,153]
	↓CCK-8, ↑Substance P → Depression	N/A	Serum	Stroke	[147]
Sex Hormones	↓Testosterone → ↓Dopamine → Anxiety	↓Testosterone is more prevalent in males	Global	TBI	[120,154]

**Table 3 cells-14-00074-t003:** Neurobiological mechanisms targeted for psychological dysfunction in preclinical brain injury models.

Mechanism	Disease/Model Implicated In	Pre-Clinical Interventional Target	References
Neurotransmission	Vascular Cognitive Impairment and Dementia	GABAr	[21,22]
		Serotonin	[102]
	Traumatic Brain Injury	GABAA	[92]
		NMDAr	[93,96,99]
		Glutamate	[95,99]
		Endocannabinoid system	[105]
	Ischemic Stroke	Dopamine	[91]
		Glutamate	[94]
		Serotonin	[103]
	Intracerebral Hemorrhage	Indoleamine 2, 3-dioxygenase and serotonin	[101]
	Subarachnoid Hemorrhage	HDAC2 and glutamate	[97]
Inflammation	Vascular Cognitive Impairment and Dementia	Angiotensin system	[128]
	Traumatic Brain Injury	Inflammatory cytokines	[72]
		Sepina3n and Neutrophil elastase	[85]
		PKA/CREB	[78]
	Ischemic Stroke	HDAC	[79]
	Intracerebral Hemorrhage	Inflammatory cytokines	[73]
		TRPV4 channel opening	[74]
Oxidative Stress	Vascular Cognitive Impairment and Dementia	Glutathione expression	[87]
		SOD	[87]
		Angiotensin system	[128]
	Traumatic Brain Injury	Lipid peroxidation	[72]
		Glutathione expression	[89,90]
	Ischemic Stroke	Lipid peroxidation	[82]
		Nrf2	[82]
	Intracerebral Hemorrhage	Nrf2	[81,83]
		SOD and ROS	[73]
Synaptic Plasticity	Vascular Cognitive Impairment and Dementia	TGF-β1	[108]
		BDNF	[21,109]
		Glycogen synthase kinase-3β	[108]
		Synaptophysin	[111]
		Lipid peroxidation	[107]
Sex Hormones	Traumatic Brain Injury	Progesterone	[119]
Stress Hormones	Traumatic Brain Injury	CRFR1 and HPA axis	[113]
	Ischemic Stroke	CRHR1	[82]
	Vascular Cognitive Impairment and Dementia	Glucocorticoid receptors	[112]
Methylation	Traumatic Brain Injury	DNMT and HDAC	[130,131]
	Ischemic Stroke	M6A demethylase and FTO	[129]
Trophic Factors	Ischemic Stroke	BDNF	[114,125]
Apoptosis	Subarachnoid Hemorrhage	p38 MAPK	[66]
Neuropeptides	Vascular Cognitive Impairment and Dementia	Angiotensin system	[128]

## Data Availability

There are no data generated for this manuscript.

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
