# Peer review of "Neurobiological Mechanisms Underlying Psychological Dysfunction After Brain Injuries"

_cells, 2025, doi:10.3390/cells14020074_

Round 1

Reviewer 1 Report

Comments and Suggestions for Authors

I appreciate the authors for presenting this review article emphasizing the neurobiological mechanisms underlying psychological dysfunction after brain injuries. This is a well-reviewed article. However, the organization needs improvement, especially regarding the figures and tables. My comments are as follows:

 1. Figure 4

This figure is not easy to read. What is the definition of "identified mechanisms which have not been targeted"? Can we find supporting evidence for this in Table 3? Please clarify.

 2. Figure 6

Do the five neuropeptides listed in the figure affect all the listed mechanisms, including neurotransmission, neuroinflammation, oxidative stress, apoptosis, synaptic plasticity, cerebral blood flow (CBF), and stress responses? Or do they only influence some of these? Please clarify.

 3. Figure 7

Do the three non-pharmacological interventions listed in the figure impact all the listed mechanisms (neurotransmission, neuroinflammation, oxidative stress, apoptosis, synaptic plasticity, CBF, and stress responses), or just some of them? Please clarify.

 4. Table 3

Please include citations for the references.

Author Response

  1. Figure 4 - This figure is not easy to read. What is the definition of "identified mechanisms which have not been targeted"? Can we find supporting evidence for this in Table 3? Please clarify.
    1. We thank the reviewer for taking the time to critically evaluate this text and provide feedback. We understand that, given the interweaving nature of the subject at hand, Figure 4 may have been difficult to read. In an effort to improve this, we have changed the colors of 4 of the mechanism labels, to increase the contrast between the text and the background color. We have also changed the color of the attendant lines as well. Additionally, the size of Figure 4 has been increased, to increase the ability to distinguish between different lines.
    2. We understand that the definition of “identified mechanisms which have not been targeted” may be somewhat unclear and have revised the figure legend to more clearly indicate that we are delineating between those mechanisms which have been directly targeted for improvement of psychological dysfunction and those which have been identified but not targeted. Furthermore, as specified in the Figure 4 legend, we have verified that each untargeted mechanism corresponds to a greyed-out entry in Table 3.
    3. We have revised the following lines in the main text:
      1. Pages 15, Lines 405-411: “While there is considerable overlap across different types of brain injury, not all mechanisms have been evaluated in every form of brain injury. Additionally, among the assessed mechanisms, only a subset has been specifically targeted with the aim of influencing psychological function (Solid, double-headed arrows = identified mechanisms which have been targeted, others have not been targeted at all for therapeutic intervention (dashed, single-headed arrows (see: Table 3, greyed out rows)).”
    4. Figure 6 - Do the five neuropeptides listed in the figure affect all the listed mechanisms, including neurotransmission, neuroinflammation, oxidative stress, apoptosis, synaptic plasticity, cerebral blood flow (CBF), and stress responses? Or do they only influence some of these? Please clarify.
      1. We recognize that the extensive number of mechanisms presented in Figure 6, alongside the associated neuropeptides, may make the figure challenging to interpret. To clarify, each mechanism listed in Figure 6 has been targeted by at least one of the neuropeptides identified. To facilitate distinction among the neuropeptides, unique icons have been assigned to each and placed adjacent to the relevant mechanisms, indicating which neuropeptide has been observed to influence specific mechanisms. The figure legend has been updated to reflect these modifications for improved clarity.
      2. We have revised the following lines in the main text:
        1. Page 24, Lines 759-767: “Neuropeptides function as multifaceted molecules that regulate several processes within the nervous system, including neurotransmission, neuroinflammation, oxidative stress, apoptosis, synaptic plasticity, cerebral blood flow (CBF), and stress responses. These pathways are integral to the development of injury-dependent psychological dysfunction, highlighting the significant potential of neuropeptides as a target for therapeutic intervention. (Each mechanism is annotated with the corresponding neuropeptides that influence it, represented by specific icons: CGRP (square), angiotensin (circle), NPY (pentagon), substance P (star), and CCK (triangle).)”
      3. Figure 7 - Do the three non-pharmacological interventions listed in the figure impact all the listed mechanisms (neurotransmission, neuroinflammation, oxidative stress, apoptosis, synaptic plasticity, CBF, and stress responses), or just some of them? Please clarify.
        1. We acknowledge that the extensive list of mechanisms and non-pharmacological interventions presented in Figure 7 may make the figure challenging to interpret. To clarify, each mechanism in the figure is targeted by at least one of the listed interventions. To facilitate distinction among the interventions, specific icons have been assigned to each and placed next to the corresponding mechanisms, indicating which intervention has been observed to target which mechanisms (Page 25). The figure legend has been revised accordingly.
        2. We have revised the following lines in the main text:
          1. Page 25, Lines 788-796: “Non-pharmacological interventions are inherently multi-mechanistic, enabling them to influence a broad array of factors that contribute to the onset of psychological dysfunction following brain injury. These factors include neurotransmission, neuroinflammation, oxidative stress, apoptosis, synaptic plasticity, cerebral blood flow (CBF), and stress responses, thereby targeting a significant portion of the pathways involved in the development of psychological distress. (Each mechanism is annotated with the interventions known to influence it, represented by specific icons: physical exercise (square), neuromodulation (circle), and an enriched environment ( triangle).)”
        3. Table 3 - Please include citations for the references.
          1. An additional column has been incorporated into Table 3 to include the corresponding references. (Pages 16-17)
  2.  

Reviewer 2 Report

Comments and Suggestions for Authors

Neurobiological Mechanisms Underlying Psychological Dysfunction After Brain Injuries

I have found the manuscript interesting, I suggest add a list of the used acronyms at end of the paper, in order to facilitate the reading.

You can find my appraisal as follows:

Introduction: In the introduction, you highlighted the role played by anxiety and depression in brain-injured patients. This is a good point and I have appreciated the section. However, I suggest writing in a better way the first sentence Psychological dysfunction, commonly manifesting as anxiety and depression, imposes a substantial global health burden”. I suppose that you can avoid introducing anxiety and depression as psychological dysfunctions. Or at least be more specific about what you mean with “psychological dysfunction” (or substitute it with distress or psychiatric symptoms/comorbidities). Please, revise.

Moreover, a critical view is good in a review paper. However, I suggest that the lines between 64-and 76 contain future direction. In this way, this section needs to be limited.

The aims are clear and well stated at the end of the section.

I like Fig. 2 but you need to disclose the acronyms in the caption.

2. Neurobiological mechanisms and therapeutic strategies for psychological dysfunction in injury-independent and injury-dependent contexts: The section is interesting. However, you need to be more specific about the role played by the subcortical regions that you mentioned play a relevant role in psychological distress and psychiatric symptoms/disorders associated with brain injuries. However, the remaining subsections of “Traumatic brain injuryare well-written. Tables 1 and 2 need abbreviations disclosed in the caption. In line 302: what do you mean by “combat”? please clarify and revise. This statement is not clear “of individuals may be diagnosed with a DSM-IV Axis I disorder, compared to 5-15% in the general population”. I suggest to be more specific.

Ischemic stroke: The subsection is well-written and interesting.

Intracerebral hemorrhage: The acronym ICH needs to be disclosed, similarly for the subsequent subsections.

The remaining subsections are clear.

3. Sexually dimorphic responses reveal potential targets for therapeutic intervention in psychological dysfunction within injured brains: The statement in lines 640-641 needs to be explicated in a better way. Specifically, this come from epidemiology or neurobiology? Please, add more info about non-injured women.

4. Future directions for enhancing therapeutic efficacy in injury-dependent psychological dysfunction: Despite the complexity of the main theme, I found it intriguing and you gave a series of future directions to stimulate further studies. However, it is not clear (923) is you mean animal models.

Author Response

Reviewer 2:

I have found the manuscript interesting, I suggest add a list of the used acronyms at end of the paper, in order to facilitate the reading.

You can find my appraisal as follows:

We appreciate the reviewer’s thorough evaluation of this review and their insightful feedback. In accordance with their recommendation, we have included an abbreviations list at the end of the manuscript, encompassing those utilized throughout the text, figures, and tables (Pages 29-32).

  1. Introduction:
    1. In the introduction, you highlighted the role played by anxiety and depression in brain-injured patients. This is a good point and I have appreciated the section. However, I suggest writing in a better way the first sentence “Psychological dysfunction, commonly manifesting as anxiety and depression, imposes a substantial global health burden”. I suppose that you can avoid introducing anxiety and depression as psychological dysfunctions. Or at least be more specific about what you mean with “psychological dysfunction” (or substitute it with distress or psychiatric symptoms/comorbidities). Please, revise.
      1. We acknowledge the reviewer’s concern regarding the potential lack of clarity in the first sentence of the introduction, particularly due to the emphasis on anxiety and depression. To address this, we have revised the sentence by removing the specific mention of anxiety and depression and instead referring to psychological dysfunction as a "spectrum of affective disorders." Anxiety and depression are now introduced in the first sentence of the second paragraph, where the focus shifts to their relationship with acquired brain injury.
      2. We have revised the following sentences in the main text.
        1. Page 1, Lines 40-41: “Psychological dysfunction, manifesting as a complex spectrum of affective disorders, imposes a substantial global health burden [1–5].
        2. Page 2, Lines 59-62: “Two of the most common forms of affective disorder, anxiety and depression [1–5], are exacerbated both by brain injury and the psycho-emotional disturbances related to the acute phase of the inciting event [13,14].”
      3. Moreover, a critical view is good in a review paper. However, I suggest that the lines between 64-and 76 contain future direction. In this way, this section needs to be limited.
        1. We agree with the reviewer that the inclusion of future directions in the introduction should be limited.
        2. However, we believe that the section in question is a concise summary of our findings rather than a discussion of future directions. Nevertheless, we agree that it should be brief in the introduction and have therefore removed part of the section.
  • We have revised the following sentences in the main text.
    1. Page 2, Lines 65-74: “Meta-analyses on injury-dependent psychological dysfunction reveal a broad range of symptom severity, from mild to severe, regardless of the initial injury severity, influenced by various contributing factors. Research into injury-dependent psychological dysfunction has often mirrored studies on injury-independent psychological dysfunction, potentially overlooking etiologically specific factors that may contribute to reduced treatment efficacy. This gap highlights the need for more effective strategies to manage established psychological dysfunction following brain injury. It is crucial to re-evaluate the current neurobiological mechanisms targeted for therapeutic interventions addressing psychological dysfunction in various brain injury models, to develop more effective treatment strategies that enhance clinical outcomes.”
  1. The aims are clear and well stated at the end of the section.
    1. We thank the reviewer for this comment.
  2. I like Fig. 2 but you need to disclose the acronyms in the caption.
    1. We appreciate the reviewer’s positive feedback on Figure 2. In response, we have included the abbreviations in the figure caption for Figure 2.
    2. We have revised the following sentences in the main text.
      1. Page 3, Lines 96-102: “Figure 2. Different types of brain injury are associated with varying prevalences of psychological dysfunction. Psychological dysfunction is more prevalent in populations with brain injury, with each injury type exhibiting distinct rates and prevalences of psychological symptoms. Notably, vascular cognitive impairment and dementia (VCID), despite showing minimal overt damage (highlighted in red), exhibit the highest prevalence of psychological dysfunction. (Created with Biorender.com) (ICH: intracerebral hemorrhage; SAH: subarachnoid hemorrhage; TBI: traumatic brain injury)”

  1. Neurobiological mechanisms and therapeutic strategies for psychological dysfunction in injury-independent and injury-dependent contexts:
    1. The section is interesting. However, you need to be more specific about the role played by the subcortical regions that you mentioned play a relevant role in psychological distress and psychiatric symptoms/disorders associated with brain injuries. However, the remaining subsections of “Traumatic brain injury” are well-written.
      1. We appreciate the reviewer’s positive comment on this section. However, as section 2.1.2 aims to establish a generalized framework for identifying brain regions involved in psychological dysfunction in non-injured contexts, we believe the level of specificity provided is appropriate. Subsequent sections, which address injured contexts, offer more detailed mechanistic insights into regional contributions. While there are numerous reviews and articles examining the specific regional and subcortical contributions to psychological distress in non-injured populations, this is not the primary focus of the current manuscript. Including additional details at this stage may distract from the later sections and the overall narrative of the manuscript.
    2. Tables 1 and 2 need abbreviations disclosed in the caption.
      1. We appreciate the reviewer’s suggestion to provide a description of the abbreviations in the table. Given the large number of abbreviations used, especially those that overlap between tables, we have included a master list of abbreviations at the end of the manuscript (Pages 29-32). This approach prevents the table captions from becoming overly lengthy. The list includes all abbreviations used in the main text, tables, and figures, for ease of reference.
    3. In line 302: what do you mean by “combat”? please clarify and revise.
      1. To clarify, by “combat” we meant military combat. This has been indicated in the text (line 304).
      2. We have revised the following sentences in the main text.
        1. Page 12, Lines 303-305: “TBI affects over 25 million individuals annually [132], with its causes encompassing a wide range of events, including military combat, vehicular accidents, recreational or sports-related injuries, and falls, although mild TBI (mTBI) is often underreported [133].”
      3. This statement is not clear “of individuals may be diagnosed with a DSM-IV Axis I disorder, compared to 5-15% in the general population”. I suggest to be more specific.
        1. We acknowledge that the original phrasing of the statement may have been unclear. Therefore, we have revised the statement to ensure greater precision and clarity.
        2. We have revised the following sentences in the main text.
          1. Page 12, Lines 311-314: “In the absence of injury, approximately 5-15% of the general population is diagnosed with a DSM-IV Axis I disorder; however, within the first year following TBI, this rate increases to as high as 60% of individuals [1–5].”
        3. Ischemic stroke: The subsection is well-written and interesting.
          1. We thank the reviewer for this comment.
        4. Intracerebral hemorrhage: The acronym ICH needs to be disclosed, similarly for the subsequent subsections.
          1. We thank the reviewer for bringing up the lack of abbreviation distinction for ICH. Accordingly, we have added to the Introduction section, alongside the other injury-type abbreviations. We have also included a list of abbreviations at the end of the manuscript, detailing those used within the text, figures, and tables (Pages 29-32).
          2. We have revised the following sentences in the main text.
            1. Page 2, Lines 78-83: “It assesses psychological dysfunction arising from ischemic stroke [8,10], intracerebral hemorrhage (ICH ) [8], subarachnoid hemorrhage (SAH) [7], traumatic brain injury (TBI) [6,9], and vascular cognitive impairment and dementia (VCID) due to chronic cerebral hypoperfusion (CCH) or as a long-term consequence of acute brain injury [25], as well as distress observed in non-injury populations”
          3. The remaining subsections are clear.
            1. We thank the reviewer for this comment.

  1. Sexually dimorphic responses reveal potential targets for therapeutic intervention in psychological dysfunction within injured brains: The statement in lines 640-641 needs to be explicated in a better way. Specifically, this come from epidemiology or neurobiology? Please, add more info about non-injured women.
    1. We agree with the reviewer that citing the epidemiological source enhances the clarity of the statement. Accordingly, we have made the necessary adjustments. However, as the purpose of the information regarding healthy trends is to highlight differences from injured contexts, we believe that further elaboration in this section would not be beneficial. Instead, we have retained the information on healthy females in lines 654-661, which directly pertains to the proposed mechanisms.
    2. We have revised the following sentences in the main text.
      1. Page 21, Lines 646-648: “Epidemiological studies indicate that, absent of injury, the female population is more likely to develop depression, anxiety, or PTSD than males [34,180–182].”

  1. Future directions for enhancing therapeutic efficacy in injury-dependent psychological dysfunction: Despite the complexity of the main theme, I found it intriguing and you gave a series of future directions to stimulate further studies. However, it is not clear (923) is you mean animal models.
    1. We recognize that the statement may have been unclear in its reference to animal models. To address this, we have explicitly clarified that the recommendations are specifically intended for animal research.
    2. We have revised the following sentences in the main text.
      1. Page 28, Lines 932-936: “While it is understandable to initially evaluate therapeutic interventions with brain injury alone as the stressor, the absence of additional stressors in animal models represents a limitation which can be addressed via the inclusion of a chronic stressor post-injury, such as chronic unpredictable mild stress [77], or the usage of animals subjected to maternal deprivation prior to injury [226,227].”

Round 2

Reviewer 1 Report

Comments and Suggestions for Authors

The revised manuscript has got improved and replied my comments item-by-items. Only one comment: in Table 3 , some columns are missing without references. Please clarify. 

Author Response

Comments: The revised manuscript has got improved and replied my comments item-by-items. Only one comment: in Table 3, some columns are missing without references. Please clarify.

Response: We sincerely thank the reviewer for their constructive feedback, which has significantly contributed to enhancing its quality. The absence of reference citations in certain columns reflects the lack of studies investigating neurobiological mechanisms for specific brain injury models. To address this, we have removed those columns from Table 3, ensuring that it only includes mechanisms that have been studied.

Reviewer 2 Report

Comments and Suggestions for Authors

The authors have addressed my concerns

Author Response

Comments: The authors have addressed my concerns.

Response: We appreciate the reviewer for dedicating their time to reviewing this manuscript and for offering valuable comments.